# WHEN VISION TRANSFORMERS OUTPERFORM RESNETS WITHOUT PRE-TRAINING OR STRONG DATA AUGMENTATIONS

**Xiangning Chen[1,2]\*,  Cho-Jui Hsieh[2],  Boqing Gong[1]**
[1]Google Research,   [2]Department of Computer Science, UCLA
`{xiangningc, bgong}@google.com, chohsieh@cs.ucla.edu`

## ABSTRACT

Vision Transformers (ViTs) and MLPs signal further efforts on replacing hand-wired features or inductive biases with general-purpose neural architectures. Existing works empower the models by massive data, such as large-scale pre-training and/or repeated strong data augmentations, and still report optimization-related problems (e.g., sensitivity to initialization and learning rates). Hence, this paper investigates ViTs and MLP-Mixers from the lens of loss geometry, intending to improve the models' data efficiency at training and generalization at inference. Visualization and Hessian reveal extremely sharp local minima of converged models. By promoting smoothness with a recently proposed sharpness-aware optimizer, we substantially improve the accuracy and robustness of ViTs and MLP-Mixers on various tasks spanning supervised, adversarial, contrastive, and transfer learning (e.g., +5.3% and +11.0% top-1 accuracy on ImageNet for ViT-B/16 and Mixer-B/16, respectively, with the simple Inception-style preprocessing). We show that the improved smoothness attributes to sparser active neurons in the first few layers. The resultant ViTs outperform ResNets of similar size and throughput when trained from scratch on ImageNet without large-scale pre-training or strong data augmentations. Model checkpoints are available at `https://github.com/google-research/vision_transformer`.

## 1 INTRODUCTION

Transformers (Vaswani et al., 2017) have become the de-facto model of choice in natural language processing (NLP) (Devlin et al., 2018; Radford et al., 2018). In computer vision, there has recently been a surge of interest in end-to-end Transformers (Dosovitskiy et al., 2021; Touvron et al., 2021b; Liu et al., 2021b; Fan et al., 2021; Arnab et al., 2021; Bertasius et al., 2021; Akbari et al., 2021) and MLPs (Tolstikhin et al., 2021; Touvron et al., 2021a; Liu et al., 2021a; Melas-Kyriazi, 2021), prompting the efforts to replace hand-wired features or inductive biases with general-purpose neural architectures powered by data-driven training. We envision these efforts may lead to a unified knowledge base that produces versatile representations for different data modalities, simplifying the inference and deployment of deep learning models in various application scenarios.

Despite the appealing potential of moving toward general-purpose neural architectures, the lack of convolution-like inductive biases also challenges the training of vision Transformers (ViTs) and MLPs. When trained on ImageNet (Deng et al., 2009) with the conventional Inception-style data preprocessing (Szegedy et al., 2016), Transformers *"yield modest accuracies of a few percentage points below ResNets of comparable size"* (Dosovitskiy et al., 2021). To boost the performance, existing works resort to large-scale pre-training (Dosovitskiy et al., 2021; Arnab et al., 2021; Akbari et al., 2021) and repeated strong data augmentations (Touvron et al., 2021b), resulting in excessive demands of data, computing, and sophisticated tuning of many hyperparameters. For instance, Dosovitskiy et al. (Dosovitskiy et al., 2021) pre-train ViTs using 304M labeled images, and Touvron et al. (2021b) repeatedly stack four strong image augmentations.

---

\*Work done as a student researcher at Google.

In this paper, we show ViTs can outperform ResNets (He et al., 2016) of even bigger sizes in both accuracy and various forms of robustness by using a principled optimizer, without the need for large-scale pre-training or strong data augmentations. MLP-Mixers (Tolstikhin et al., 2021) also become on par with ResNets.

We first study the architectures fully trained on ImageNet from the lens of loss landscapes and draw the following findings. First, visualization and Hessian matrices of the loss landscapes reveal that Transformers and MLP-Mixers converge at extremely sharp local minima, whose largest principal curvatures are almost an order of magnitude bigger than ResNets'. Such effect accumulates when the gradients backpropagate from the last layer to the first, and the initial embedding layer suffers the largest eigenvalue of the corresponding sub-diagonal Hessian. Second, the networks all have very small training errors, and MLP-Mixers are more prone to overfitting than ViTs of more parameters (because of the difference in self-attention). Third, ViTs and MLP-Mixers have worse "trainabilities" than ResNets following the neural tangent kernel analyses (Xiao et al., 2020).

Therefore, we need improved learning algorithms to prevent the convergence to a sharp local minimum when it comes to the convolution-free ViTs and MLP-Mixers. The first-order optimizers (e.g., SGD and Adam (Kingma & Ba, 2015)) only seek the model parameters that minimize the training error. They dismiss the higher-order information such as flatness that correlates with generalization (Keskar et al., 2017; Kleinberg et al., 2018; Jastrzębski et al., 2019; Smith & Le, 2018; Chaudhari et al., 2017).

The above study and reasoning lead us to the recently proposed sharpness-aware minimizer (SAM) (Foret et al., 2021) that explicitly smooths the loss geometry during model training. SAM strives to find a solution whose entire neighborhood has low losses rather than focus on any singleton point. We show that the resultant models exhibit smoother loss landscapes, and their generalization capabilities improve tremendously across different tasks including supervised, adversarial, contrastive, and transfer learning (e.g., +5.3% and +11.0% top-1 accuracy on ImageNet for ViT-B/16 and Mixer-B/16, respectively, with the simple Inception-style preprocessing). The enhanced ViTs achieve better accuracy and robustness than ResNets of similar and bigger sizes when trained from scratch on ImageNet, without large-scale pre-training or strong data augmentations. Moreover, we demonstrate that SAM can even enable ViT to be effectively trained with (momentum) SGD, which usually lies far behind Adam when training Transformers (Zhang et al., 2020).

By analyzing some intrinsic model properties, we observe that SAM increases the sparsity of active neurons (especially for the first few layers), which contribute to the reduced Hessian eigenvalues. The weight norms increase, implying the commonly used weight decay may not be an effective regularization alone. A side observation is that, unlike ResNets and MLP-Mixers, ViTs have extremely sparse active neurons (see Figure 2 (right)), revealing the potential for network pruning (Akbari et al., 2021). Another interesting finding is that the improved ViTs appear to have visually more interpretable attention maps. Finally, we draw similarities between SAM and strong augmentations (e.g., mixup) in that they both smooth the average loss geometry and encourage the models to behave linearly between training images.

## 2 BACKGROUND AND RELATED WORK

We briefly review ViTs, MLP-Mixers, and some related works in this section.

Dosovitskiy et al. (2021) show that a pure Transformer architecture (Vaswani et al., 2017) can achieve state-of-the-art accuracy on image classification by pre-training it on large datasets such as ImageNet-21k (Deng et al., 2009) and JFT-300M (Sun et al., 2017). Their vision Transformer (ViT) is a stack of residual blocks, each containing a multi-head self-attention, layer normalization (Ba et al., 2016), and a MLP layer. ViT first embeds an input image $x \in \mathbb{R}^{H \times W \times C}$ into a sequence of features $z \in \mathbb{R}^{N \times D}$ by applying a linear projection over $N$ nonoverlapping image patches $x_p \in \mathbb{R}^{N \times (P^2 \cdot C)}$, where $D$ is the feature dimension, $P$ is the patch resolution, and $N = HW/P^2$ is the sequence length. The self-attention layers in ViT are global and do not possess the locality and translation equivariance of convolutions. ViT is compatible with the popular architectures in NLP (Devlin et al., 2018; Radford et al., 2018) and, similar to its NLP counterparts, requires pre-training over massive datasets (Dosovitskiy et al., 2021; Akbari et al., 2021; Arnab et al., 2021) or

Table 1: Number of parameters, NTK condition number $\kappa$, Hessian dominate eigenvalue $\lambda_{max}$, training error at convergence $L_{train}$, average flatness $L_{train}^{\mathcal{N}}$, accuracy on ImageNet, and accuracy/robustness on ImageNet-C. ViT and MLP-Mixer suffer divergent $\kappa$ and converge at sharp regions; SAM rescues that and leads to better generalization.

| | ResNet-152 | ResNet-152-SAM | ViT-B/16 | ViT-B/16-SAM | Mixer-B/16 | Mixer-B/16-SAM |
|---|---|---|---|---|---|---|
| **#Params** | 60M | | 87M | | 59M | |
| **NTK** $\kappa$ [†] | 2801.6 | | 4205.3 | | 14468.0 | |
| **Hessian** $\lambda_{max}$ | 179.8 | **42.0** | 738.8 | **20.9** | 1644.4 | **22.5** |
| $L_{train}$ | **0.86** | 0.90 | **0.65** | 0.82 | **0.45** | 0.97 |
| $L_{train}^{\mathcal{N}}$ [⋆] | 2.39 | **2.16** | 6.66 | **0.96** | 7.78 | **1.01** |
| **ImageNet (%)** | 78.5 | **79.3** | 74.6 | **79.9** | 66.4 | **77.4** |
| **ImageNet-C (%)** | 50.0 | **52.2** | 46.6 | **56.5** | 33.8 | **48.8** |

[†] As it is prohibitive to compute the exact NTK, we approximate the value by averaging over its sub-diagonal blocks (see Appendix G for details). We average the results for 1,000 random noises when calculating $L_{train}^{\mathcal{N}}$.

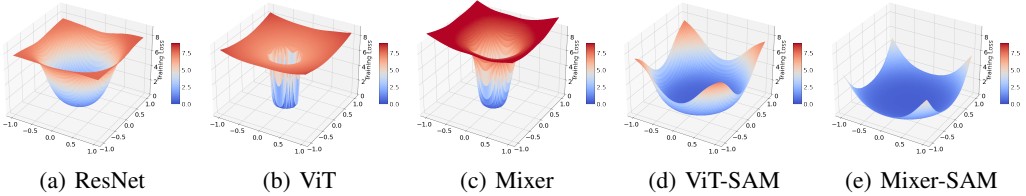

(a) ResNet  (b) ViT  (c) Mixer  (d) ViT-SAM  (e) Mixer-SAM

Figure 1: Cross-entropy loss landscapes of ResNet-152, ViT-B/16, and Mixer-B/16. ViT and MLP-Mixer converge to sharper regions than ResNet when trained on ImageNet with the basic Inception-style preprocessing. SAM, a sharpness-aware optimizer, significantly smooths the landscapes.

strong data augmentations (Touvron et al., 2021b). Some works specialize the ViT architectures for visual data (Liu et al., 2021b; Yuan et al., 2021; Fan et al., 2021; Bertasius et al., 2021).

More recent works find that the self-attention in ViT is not vital for performance, resulting in several architectures exclusively based on MLPs (Tolstikhin et al., 2021; Touvron et al., 2021a; Liu et al., 2021a; Melas-Kyriazi, 2021). Here we take MLP-Mixer (Tolstikhin et al., 2021) as an example. MLP-Mixer shares the same input layer as ViT; namely, it partitions an image into a sequence of nonoverlapping patches/tokens. It then alternates between token and channel MLPs, where the former allows feature fusion from different spatial locations.

We focus on ViTs and MLP-Mixers in this paper. We denote by "S" and "B" the small and base model sizes, respectively, and by an integer the image patch resolution. For instance, ViT-B/16 is the base ViT model taking as input a sequence of $16 \times 16$ patches. Appendices contain more details.

## 3 ViTs and MLP-Mixers Converge at Sharp Local Minima

The current training recipe of ViTs, MLP-Mixers, and related convolution-free architectures relies heavily on massive pre-training (Dosovitskiy et al., 2021; Arnab et al., 2021; Akbari et al., 2021) or a bag of strong data augmentations (Touvron et al., 2021b; Tolstikhin et al., 2021; Cubuk et al., 2019; 2020; Zhang et al., 2018; Yun et al., 2019). It highly demands data and computing, and leads to many hyperparameters to tune. Existing works report that ViTs yield inferior accuracy to the ConvNets of similar size and throughput when trained from scratch on ImageNet without the combination of those advanced data augmentations, despite using various regularization techniques (e.g., large weight decay, Dropout (Srivastava et al., 2014), etc.). For instance, ViT-B/16 (Dosovitskiy et al., 2021) gives rise to 74.6% top-1 accuracy on the ImageNet validation set (224 image resolution), compared with 78.5% of ResNet-152 (He et al., 2016). Mixer-B/16 (Tolstikhin et al., 2021) performs even worse (66.4%). There also exists a large gap between ViTs and ResNets in robustness tests (see Table 2 for details).

Moreover, Chen et al. (2021c) find that the gradients can spike and cause a sudden accuracy dip when training ViTs, and Touvron et al. (2021b) report the training is sensitive to initialization and hyperparameters. These all point to optimization problems. In this paper, we investigate the loss

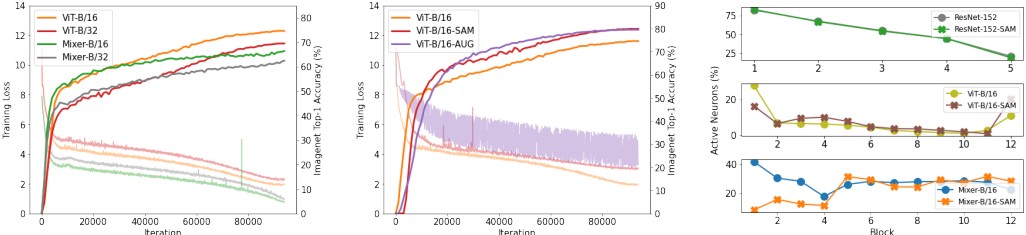

Figure 2: **Left** and **Middle**: ImageNet training error and validation accuracy vs. iteration for ViTs and MLP-Mixers. **Right**: Percentage of active neurons for ResNet-152, ViT-B/16, and Mixer-B/16.

landscapes of ViTs and MLP-Mixers to understand them from the optimization perspective, intending to reduce their dependency on the large-scale pre-training or strong data augmentations.

**ViTs and MLP-Mixers converge at extremely sharp local minima.** It has been extensively studied that the convergence to a flat region whose curvature is small benefits the generalization of neural networks (Keskar et al., 2017; Kleinberg et al., 2018; Jastrzębski et al., 2019; Chen & Hsieh, 2020; Smith & Le, 2018; Zela et al., 2020; Chaudhari et al., 2017). Following Li et al. (2018), we plot the loss landscapes at convergence when ResNets, ViTs, and MLP-Mixers are trained from scratch on ImageNet with the basic Inception-style preprocessing (Szegedy et al., 2016) (see Appendices for details). As shown in Figures 1(a) to 1(c), ViTs and MLP-Mixers converge at much sharper regions than ResNets. Besides, we calculate the training error under Gaussian perturbations on the model parameters $L_{train}^{\mathcal{N}} = \mathbb{E}_{\epsilon \sim \mathcal{N}}[L_{train}(w + \epsilon)]$ in Table 1, which reveals the *average* flatness. Although ViT-B/16 and Mixer-B/16 achieve lower training error $L_{train}$ than that of ResNet-152, their loss values after random weight perturbation become much higher. We further validate the results by computing the dominate Hessian eigenvalue $\lambda_{max}$, which is a mathematical evaluation of the *worst-case* landscape curvature. The $\lambda_{max}$ values of ViT and MLP-Mixer are orders of magnitude larger than that of ResNet, and MLP-Mixer suffers the largest curvature among the three species (see Section 4.4 for a detailed analysis).

**Small training errors.** This convergence at sharp regions coincides with the training dynamics shown in Figure 2 (left). Although Mixer-B/16 has fewer parameters than ViT-B/16 (59M vs. 87M), it has a smaller training error (also see $L_{train}$ in Table 1) but much worse test accuracy, implying that using the cross-token MLP to learn the interplay across image patches is more prone to overfitting than ViTs' self-attention mechanism whose behavior is restricted by a softmax. To validate this statement, we simply remove the softmax in ViT-B/16, such that the query and key matrices can freely interact with each other. Although having lower $L_{train}$ (0.56 vs. 0.65), the obtained ViT-B/16-Free performs much worse than the original ViT-B/16 (70.5% vs. 74.6%). Its $L_{train}^{\mathcal{N}}$ and $\lambda_{max}$ are 7.01 and 1236.2, revealing that ViT-B/16-Free converges to a sharper region than ViT-B/16 ($L_{train}^{\mathcal{N}}$ is 6.66 and $\lambda_{max}$ is 738.8) both on average and in the worst-case direction. Such a difference probably explains why it is easier for MLP-Mixers to get stuck in sharp local minima.

**ViTs and MLP-Mixers have worse trainability.** Furthermore, we discover that ViTs and MLP-Mixers suffer poor trainabilities, defined as the effectiveness of a network to be optimized by gradient descent (Xiao et al., 2020; Burkholz & Dubatovka, 2019; Shin & Karniadakis, 2020). Xiao et al. (2020) show that the trainability of a neural network can be characterized by the condition number of the associated neural tangent kernel (NTK), $\Theta(x, x') = J(x)J(x')^T$, where $J$ is the Jacobian matrix. Denoting by $\lambda_1 \geq \cdots \geq \lambda_m$ the eigenvalues of NTK $\Theta_{train}$, the smallest eigenvalue $\lambda_m$ converges exponentially at a rate given by the condition number $\kappa = \lambda_1 / \lambda_m$. If $\kappa$ diverges then the network will become untrainable (Xiao et al., 2020; Chen et al., 2021a). As shown in Table 1, $\kappa$ is pretty stable for ResNets, echoing previous results that ResNets enjoy superior trainability regardless of the depth (Yang & Schoenholz, 2017; Li et al., 2018). However, we observe that the condition number diverges when it comes to ViT and MLP-Mixer, confirming that the training of ViTs desires extra care (Chen et al., 2021c; Touvron et al., 2021b).

## 4 A PRINCIPLED OPTIMIZER FOR CONVOLUTION-FREE ARCHITECTURES

The commonly used first-order optimizers (e.g., SGD (Nesterov, 1983), Adam (Kingma & Ba, 2015)) only seek to minimize the training loss $L_{train}(w)$. They usually dismiss the higher-order

information such as curvature that correlates with the generalization (Keskar et al., 2017; Chaudhari et al., 2017; Dziugaite & Roy, 2017). However, the objective $L_{train}$ for deep neural networks are highly non-convex, making it easy to reach near-zero training error but high generalization error $L_{test}$ during evaluation, let alone their robustness when the test sets have different distributions (Hendrycks & Dietterich, 2019; Hendrycks et al., 2020). ViTs and MLPs amplify such drawbacks of first-order optimizers due to the lack of inductive bias for visual data, resulting in excessively sharp loss landscapes and poor generalization, as shown in the previous section. We hypothesize that smoothing the loss landscapes at convergence can significantly improve the generalization ability of those convolution-free architectures, leading us to the recently proposed sharpness-aware minimizer (SAM) (Foret et al., 2021) that explicitly avoids sharp minima.

### 4.1 SAM: Overview

Intuitively, SAM (Foret et al., 2021) seeks to find the parameter $w$ whose entire neighbours have low training loss $L_{train}$ by formulating a minimax objective:

$$\min_{w} \max_{\|\epsilon\|_2 \le \rho} L_{train}(w + \epsilon), \tag{1}$$

where $\rho$ is the size of the neighbourhood ball. Without loss of generality, here we use $l_2$ norm for its strong empirical results (Foret et al., 2021) and omit the regularization term for simplicity. Since the exact solution of the inner maximization $\epsilon^{\star} = \arg\max_{\|\epsilon\|_2 \le \rho} L_{train}(w + \epsilon)$ is hard to obtain, they employ an efficient first-order approximation:

$$\hat{\epsilon}(w) = \arg\max_{\|\epsilon\|_2 \le \rho} L_{train}(w) + \epsilon^T \nabla_w L_{train}(w) = \rho \nabla_w L_{train}(w) / \|\nabla_w L_{train}(w)\|_2. \tag{2}$$

Under the $l_2$ norm, $\hat{\epsilon}(w)$ is simply a scaled gradient of the current weight $w$. After computing $\hat{\epsilon}$, SAM updates $w$ based on the sharpness-aware gradient $\nabla_w L_{train}(w)|_{w+\hat{\epsilon}(w)}$.

### 4.2 Sharpness-aware optimization improves ViTs and MLP-Mixers

We train ViTs and MLP-Mixers with no large-scale pre-training or strong data augmentations. We directly apply SAM to the original ImageNet training pipeline of ViTs (Dosovitskiy et al., 2021) without changing any hyperparameters. The pipeline employs the basic Inception-style preprocessing (Szegedy et al., 2016). The original training setup of MLP-Mixers (Tolstikhin et al., 2021) includes a combination of strong data augmentations, and we replace it with the same Inception-style preprocessing for a fair comparison. Note that we perform grid search for the learning rate, weight decay, Dropout *before* applying SAM. Please see Appendices for training details.

**Smoother regions around the local minima.** Thanks to SAM, both ViTs and MLP-Mixers converge at much smoother regions, as shown in Figures 1(d) and 1(e). Moreover, both the average and the worst-case curvature, i.e., $L_{train}^{\mathcal{N}}$ and $\lambda_{max}$, decrease dramatically (see Table 1).

**Higher accuracy.** What comes along is tremendously improved generalization performance. On ImageNet, SAM boosts the top-1 accuracy of ViT-B/16 from 74.6% to 79.9%, and Mixer-B/16 from 66.4% to 77.4%. For comparison, the improvement on a similarly sized ResNet-152 is 0.8%. Empirically, *the degree of improvement negatively correlates with the constraints of inductive biases built into the architecture.* ResNets with inherent translation equivalence and locality benefit less from landscape smoothing than the attention-based ViTs. MLP-Mixers gain the most from the smoothed loss geometry. In Table 3, we further train two hybrid models (Dosovitskiy et al., 2021) to validate this observation, where the Transformer takes the feature map extracted from a ResNet-50 as the input sequence. The improvement brought by SAM decreases after we introduce the convolution to ViT, for instance, +2.7% for R50-B/16 compared to +5.3% for ViT-B/16. Moreover, SAM brings larger improvements to the models of larger capacity (e.g., +4.1% for Mixer-S/16 vs. +11.0% for Mixer-B/16) and longer patch sequence (e.g., +2.1% for ViT-S/32 vs. +5.3% for ViT-S/8). Please see Table 2 for more results.

SAM can be easily applied to common base optimizers. Besides Adam, we also apply SAM on top of the (momentum) SGD that usually performs much worse than Adam when training Transformers (Zhang et al., 2020). As expected, we find that under the same training budget (300 epochs), the ViT-B/16 trained with SGD only achieves 71.5% accuracy on ImageNet, whereas Adam achieves

Table 2: Performance of ResNets, ViTs, and MLP-Mixers trained from scratch on ImageNet with SAM (improvement over the vanilla model is shown in the parentheses). We use the Inception-style preprocessing (with resolution 224) rather than a combination of strong data augmentations.

| Model | #params | Throughput (img/sec/core) | ImageNet | ReaL | V2 | ImageNet-R | ImageNet-C |
|---|---|---|---|---|---|---|---|
| **ResNet** | | | | | | | |
| ResNet-50-SAM | 25M | 2161 | 76.7 (+0.7) | 83.1 (+0.7) | 64.6 (+1.0) | 23.3 (+1.1) | 46.5 (+1.9) |
| ResNet-101-SAM | 44M | 1334 | 78.6 (+0.8) | 84.8 (+0.9) | 66.7 (+1.4) | 25.9 (+1.5) | 51.3 (+2.8) |
| ResNet-152-SAM | 60M | 935 | 79.3 (+0.8) | 84.9 (+0.7) | 67.3 (+1.0) | 25.7 (+0.4) | 52.2 (+2.2) |
| ResNet-50x2-SAM | 98M | 891 | 79.6 (+1.5) | 85.3 (+1.6) | 67.5 (+1.7) | 26.0 (+2.9) | 50.7 (+3.9) |
| ResNet-101x2-SAM | 173M | 519 | 80.9 (+2.4) | 86.4 (+2.4) | 69.1 (+2.8) | 27.8 (+3.2) | 54.0 (+4.7) |
| ResNet-152x2-SAM | 236M | 356 | 81.1 (+1.8) | 86.4 (+1.9) | 69.6 (+2.3) | 28.1 (+2.8) | 55.0 (+4.2) |
| **Vision Transformer** | | | | | | | |
| ViT-S/32-SAM | 23M | 6888 | 70.5 (+2.1) | 77.5 (+2.3) | 56.9 (+2.6) | 21.4 (+2.4) | 46.2 (+2.9) |
| ViT-S/16-SAM | 22M | 2043 | 78.1 (+3.7) | 84.1 (+3.7) | 65.6 (+3.9) | 24.7 (+4.7) | 53.0 (+6.5) |
| ViT-S/14-SAM | 22M | 1234 | 78.8 (+4.0) | 84.8 (+4.5) | 67.2 (+5.2) | 24.4 (+4.7) | 54.2 (+7.0) |
| ViT-S/8-SAM | 22M | 333 | 81.3 (+5.3) | 86.7 (+5.5) | 70.4 (+6.2) | 25.3 (+6.1) | 55.6 (+8.5) |
| ViT-B/32-SAM | 88M | 2805 | 73.6 (+4.1) | 80.3 (+5.1) | 60.0 (+4.7) | 24.0 (+4.1) | 50.7 (+6.7) |
| ViT-B/16-SAM | 87M | 863 | 79.9 (+5.3) | 85.2 (+5.4) | 67.5 (+6.2) | 26.4 (+6.3) | 56.5 (+9.9) |
| **MLP-Mixer** | | | | | | | |
| Mixer-S/32-SAM | 19M | 11401 | 66.7 (+2.8) | 73.8 (+3.5) | 52.4 (+2.9) | 18.6 (+2.7) | 39.3 (+4.1) |
| Mixer-S/16-SAM | 18M | 4005 | 72.9 (+4.1) | 79.8 (+4.7) | 58.9 (+4.1) | 20.1 (+4.2) | 42.0 (+6.4) |
| Mixer-S/8-SAM | 20M | 1498 | 75.9 (+5.7) | 82.5 (+6.3) | 62.3 (+6.2) | 20.5 (+5.1) | 42.4 (+7.8) |
| Mixer-B/32-SAM | 60M | 4209 | 72.4 (+9.9) | 79.0 (+10.9) | 58.0 (+10.4) | 22.8 (+8.2) | 46.2 (12.4) |
| Mixer-B/16-SAM | 59M | 1390 | 77.4 (+11.0) | 83.5 (+11.4) | 63.9 (+13.1) | 24.7 (+10.2) | 48.8 (+15.0) |
| Mixer-B/8-SAM | 64M | 466 | 79.0 (+10.4) | 84.4 (+10.1) | 65.5 (+11.6) | 23.5 (+9.2) | 48.9 (+16.9) |

74.6%. Surprisingly, SGD + SAM can push the result to 79.1%, which is a huge +7.6% absolute improvement. Although Adam + SAM is still higher (79.9%), their gap largely shrinks.

**Better robustness.** We also evaluate the models' robustness using ImageNet-R (Hendrycks et al., 2020) and ImageNet-C (Hendrycks & Dietterich, 2019) and find even bigger impacts of the smoothed loss landscapes. On ImageNet-C, which corrupts images by noise, bad weather, blur, etc., we report the average accuracy against 19 corruptions across five levels. As shown in Tables 1 and 2, the accuracies of ViT-B/16 and Mixer-B/16 increase by 9.9% and 15.0% (which are 21.2% and 44.4% *relative* improvements), after SAM smooths their converged local regions. In comparison, SAM improves the accuracy of ResNet-152 by 2.2% (4.4% *relative* improvement). We can see that SAM enhances the robustness even more than the *relative* clean accuracy improvements (7.1%, 16.6%, and 1.0% for ViT-B/16, Mixer-B/16, and ResNet-152, respectively).

### 4.3 ViTs outperform ResNets without pre-training or strong augmentations

The performance of an architecture is often conflated with the training strategies (Bello et al., 2021), where data augmentations play a key role (Cubuk et al., 2019; 2020; Zhang et al., 2018; Xie et al., 2020; Chen et al., 2021b). However, the design of augmentations requires substantial domain expertise and may not translate between images and videos, for instance. Thanks to the principled sharpness-aware optimizer, we can remove the advanced augmentations and focus on the architectures themselves.

Table 3: Accuracy and robustness of two hybrid architectures.

| Model | #params | ImageNet (%) | ImageNet-C (%) |
|---|---|---|---|
| R50-S/16 | 34M | 79.8 | 53.4 |
| R50-S/16-SAM | | 81.0 (+1.2) | 57.2 (+3.8) |
| R50-B/16 | 99M | 79.7 | 54.4 |
| R50-B/16-SAM | | 82.4 (+2.7) | 61.0 (+6.6) |

When trained from scratch on ImageNet with SAM, *ViTs outperform ResNets of similar and greater sizes (also comparable throughput at inference)* regarding both clean accuracy (on ImageNet (Deng et al., 2009), ImageNet-ReaL (Beyer et al., 2020), and ImageNet V2 (Recht et al., 2019)) and robustness (on ImageNet-R (Hendrycks et al., 2020) and ImageNet-C (Hendrycks & Dietterich, 2019)). ViT-B/16 achieves 79.9%, 26.4%, and 56.6% top-1 accuracy on ImageNet, ImageNet-R, and ImageNet-C, while the counterpart numbers for ResNet-152 are 79.3%, 25.7%, and 52.2%, respectively (see Table 2). The gaps between ViTs and ResNets are even wider for small architectures. ViT-S/16 outperforms a similarly sized ResNet-50 by 1.4% on ImageNet, and 6.5% on ImageNet-C. SAM also significantly improves MLP-Mixers' results.

Table 4: Dominant eigenvalue $\lambda_{max}$ of the sub-diagonal Hessians for different network components, and norm of the model parameter $w$ and the post-activation $a_k$ of block $k$. Each ViT block consists of a MSA and a MLP, and MLP-Mixer alternates between a token MLP a channel MLP. Shallower layers have larger $\lambda_{max}$. SAM smooths every component.

| Model | $\lambda_{max}$ **of diagonal blocks of Hessian** | | | | | | | $\|w\|_2$ | $\|a_1\|_2$ | $\|a_6\|_2$ | $\|a_{12}\|_2$ |
|---|---|---|---|---|---|---|---|---|---|---|---|
| | Embedding | MSA/
Token MLP | MLP/
Channel MLP | Block1 | Block6 | Block12 | Whole | | | | |
| ViT-B/16 | 300.4 | 179.8 | 281.4 | 44.4 | 32.4 | 26.9 | 738.8 | 269.3 | 104.9 | 104.3 | 138.1 |
| ViT-B/16-SAM | 3.8 | 8.5 | 9.6 | 1.7 | 1.7 | 1.5 | 20.9 | 353.8 | 117.0 | 120.3 | 97.2 |
| Mixer-B/16 | 1042.3 | 95.8 | 417.9 | 239.3 | 41.2 | 5.1 | 1644.4 | 197.6 | 96.7 | 135.1 | 74.9 |
| Mixer-B/16-SAM | 18.2 | 1.4 | 9.5 | 4.0 | 1.1 | 0.3 | 22.5 | 389.9 | 110.9 | 176.0 | 216.1 |

## 4.4 INTRINSIC CHANGES AFTER SAM

We take a deeper look into the models to understand how they intrinsically change to reduce the Hessian' eigenvalue $\lambda_{max}$ and what the changes imply in addition to the enhanced generalization.

**Smoother loss landscapes for every network component.** In Table 4, we break down the Hessian of the whole architecture into small diagonal blocks of Hessians concerning each set of parameters, attempting to analyze what specific components cause the blowing up of $\lambda_{max}$ in the models trained without SAM. We observe that shallower layers have larger Hessian eigenvalues $\lambda_{max}$, and the first linear embedding layer incurs the sharpest geometry. This agrees with the finding in (Chen et al., 2021c) that spiking gradients happen early in the embedding layer. Additionally, the multi-head self-attention (MSA) in ViTs and the Token MLPs in MLP-Mixers, both of which mix information across spatial locations, have comparably lower $\lambda_{max}$ than the other network components. SAM consistently reduces the $\lambda_{max}$ of all network blocks.

We can gain insights into the above findings by the recursive formulation of Hessian matrices for MLPs (Botev et al., 2017). Let $h_k$ and $a_k$ be the pre-activation and post-activation values for layer $k$, respectively. They satisfy $h_k = W_k a_{k-1}$ and $a_k = f_k(h_k)$, where $W_k$ is the weight matrix and $f_k$ is the activation function (GELU (Hendrycks & Gimpel, 2020) in MLP-Mixers). Here we omit the bias term for simplicity. The diagonal block of Hessian matrix $H_k$ with respect to $W_k$ can be recursively calculated as:

$$H_k = (a_{k-1}a_{k-1}^T) \otimes \mathcal{H}_k, \quad \mathcal{H}_k = B_k W_{k+1}^T \mathcal{H}_{k+1} W_{k+1} B_k + D_k, \tag{3}$$

$$B_k = \text{diag}(f_k'(h_k)), \qquad D_k = \text{diag}(f_k''(h_k)\frac{\partial L}{\partial a_k}), \tag{4}$$

where $\otimes$ is the Kronecker product, $\mathcal{H}_k$ is the pre-activation Hessian for layer $k$, and $L$ is the objective function. Therefore, the Hessian norm accumulates as the recursive formulation backpropagates to shallow layers, explaining why the first block has much larger $\lambda_{max}$ than the last block in Table 4.

**Greater weight norms.** After applying SAM, we find that in most cases, the norm of the post-activation value $a_{k-1}$ and the weight $W_{k+1}$ become even bigger (see Table 4), indicating that the commonly used weight decay may not effectively regularize ViTs and MLP-Mixers (see Appendix J for further verification when we vary the weight decay strength).

**Sparser active neurons in MLP-Mixers.** Given the recursive formulation Equation (3), we identify another intrinsic measure of MLP-Mixers that contribute to the Hessian: the number of activated neurons. Indeed, $B_k$ is determined by the activated neurons whose values are greater than zero, since the first-order derivative of GELU becomes much smaller when the input is negative. As a result, the number of active GELU neurons is directly connected to the Hessian norm. Figure 2 (right) shows the proportion of activated neurons for each block, counted using 10% of the ImageNet training set. We can see that SAM greatly reduces the proportion of activated neurons for the first few layers of the Mixer-B/16, pushing them to much sparser states. This result also suggests the potential redundancy of image patches.

**ViTs' active neurons are highly sparse.** Although Equations (3) and (4) only involve MLPs, we still observe a decrease of activated neurons in the first layer of ViTs (but not as significant as in MLP-Mixers). More interestingly, we find that the proportion of active neurons in ViT is much smaller than another two architectures — given an input image, less than 10% neurons have values greater than zero for most layers (see Figure 2 (right)). In other words, ViTs offer a huge potential for

Table 5: Data augmentations, SAM, and their combination applied to different model architectures trained on ImageNet and its subsets from scratch.

| Dataset | ResNet-152 | | | | ViT-B/16 | | | | Mixer-B/16 | | | |
|---|---|---|---|---|---|---|---|---|---|---|---|---|
| | Vanilla | SAM | AUG | SAM + AUG | Vanilla | SAM | AUG | SAM + AUG | Vanilla | SAM | AUG | SAM + AUG |
| ImageNet | 78.5 | 79.3 | 78.8 | 78.9 | 74.6 | 79.9 | 79.6 | 81.5 | 66.4 | 77.4 | 76.5 | 78.1 |
| i1k (1/2) | 74.2 | 75.6 | 75.1 | 75.5 | 64.9 | 75.4 | 73.1 | 75.8 | 53.9 | 71.0 | 70.4 | 73.1 |
| i1k (1/4) | 68.0 | 70.3 | 70.2 | 70.6 | 52.4 | 66.8 | 63.2 | 65.6 | 37.2 | 62.8 | 61.0 | 65.8 |
| i1k (1/10) | 54.6 | 57.1 | 59.2 | 59.5 | 32.8 | 46.1 | 38.5 | 45.7 | 21.0 | 43.5 | 43.0 | 51.0 |

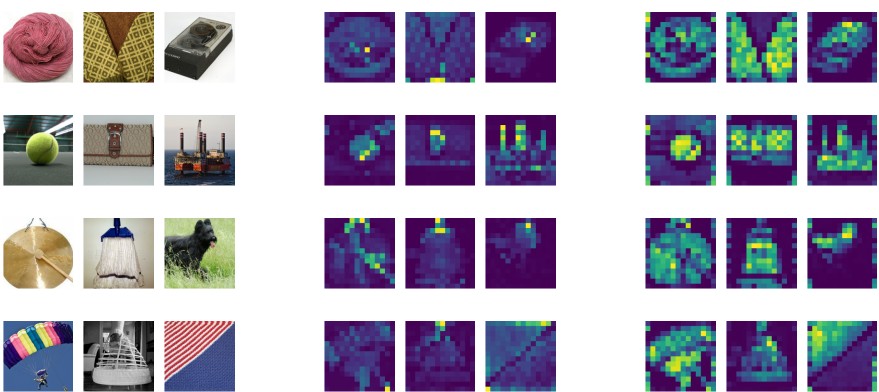

Figure 3: Raw images (**Left**) and attention maps of ViT-S/16 with (**Right**) and without (**Middle**) sharpness-aware optimization.

network pruning. This sparsity may also explain why one Transformer can handle multi-modality signals (vision, text, and audio) (Akbari et al., 2021).

**Visually improved attention maps in ViTs.** We visualize ViT-S/16's attention map of the classification token averaged over the last multi-head attentions in Figure 3 following Caron et al. (2021). Interestingly, the ViT model optimized with SAM appears to possess visually improved attention map compared with the one trained via the vanilla AdamW optimizer.

### 4.5 SAM VS. STRONG AUGMENTATIONS

Previous sections show that SAM can improve the generalization (and robustness) of ViTs and MLP-Mixers. Meanwhile, another paradigm to train these models on ImageNet from scratch is to stack multiple strong augmentations (Touvron et al., 2021b;a; Tolstikhin et al., 2021). Hence, it is interesting to study the differences and similarities between the models trained by SAM and by using strong data augmentations. For the augmentation experiments, we follow Tolstikhin et al. (2021)'s pipeline that includes mixup (Zhang et al., 2018) and RandAugment (Cubuk et al., 2020).

**Generalization.** Table 5 shows the results of strong data augmentation, SAM, and their combination on ImageNet. Each row corresponds to a training set of a different fraction of ImageNet-1k. SAM benefits ViT-B/16 and Mixer-B/16 more than the strong data augmentations, especially when the training set is small. For instance, when the training set contains only 1/10 of ImageNet training images, ViT-B/16-SAM outperforms ViT-B/16-AUG by 7.6%. Apart from the improved validation accuracy, we also observe that both SAM and strong augmentations increase the training error (see Figure 2 (Middle) and Table 6), indicating their regularization effects. However, they have distinct training dynamics as the loss curve for ViT-B/16-AUG is much nosier than ViT-B/16-SAM.

**Sharpness at convergence.** Another intriguing question is as follows. Can augmentations also smooth the loss geometry similarly to SAM? To answer it, we also plot the landscape of ViT-B/16-AUG (see Figure 5 in the Appendix) and compute its Hessian $\lambda_{max}$ together with the average flatness $L_{train}^{\mathcal{N}}$ in Table 6. Surprisingly, strong augmentations even enlarge the $\lambda_{max}$.

Table 6: Comparison between ViT-B/16-SAM and ViT-B/16-AUG. $R$ denotes the missing rate under linear interpolation.

| Model | $\lambda_{max}$ | $L_{train}$ | $L_{train}^{\mathcal{N}}$ | $R(\downarrow)$ |
|---|---|---|---|---|
| ViT-B/16 | 738.8 | 0.65 | 6.66 | 57.9% |
| ViT-B/16-SAM | 20.9 | 0.82 | 0.96 | 39.6% |
| ViT-B/16-AUG | 1659.3 | 0.85 | 1.23 | 21.4% |

However, like SAM, augmentations make ViT-B/16-AUG smoother and achieve a significantly smaller training error under random Gaussian perturbations than ViT-B/16. These results show that both SAM and augmentations make the loss landscape flat *on average*. The difference is that SAM enforces the smoothness by reducing the largest curvature via a minimax formulation to optimize the *worst-case* scenario, while augmentations ignore the worse-case curvature and instead smooth the landscape over the directions induced by the augmentations.

Interestingly, besides the similarity in smoothing the loss curvature on average, we also discover that SAM-trained models possess "linearality" resembling the property manually injected by the mixup augmentation. Following Zhang et al. (2018), we compute the prediction error in-between training data in Table 6, where a prediction $y$ is counted as a miss if it does not belong to $\{y_i, y_j\}$ evaluated at $x = 0.5x_i + 0.5x_j$. We observe that SAM greatly reduces the missing rate ($R$) compared with the vanilla baseline, showing a similar effect to mixup that explicitly encourages such linearity.

## 5 ABLATION STUDIES

In this section, we provide a more comprehensive study about SAM's effect on various vision models and under different training setups. We refer to Appendices B to D for the adversarial, contrastive and transfer learning results.

### 5.1 WHEN SCALING THE TRAINING SET SIZE

Previous studies scale up training data to show massive pre-training trumps inductive biases (Dosovitskiy et al., 2021; Tolstikhin et al., 2021). Here we show SAM further enables ViTs and MLP-Mixers to handle small-scale training data well. We randomly sample 1/4 and 1/2 images from each ImageNet class to compose two smaller-scale training sets, i.e., i1k (1/4) and i1k (1/2) with 320,291 and 640,583 images, respectively. We also use ImageNet-21k to pre-train the models with SAM, followed by fine-tuning on ImageNet-1k without SAM. The ImageNet validation set remains intact. SAM can still bring improvement when pre-trained on ImageNet-21k (+0.3%, +1.4%, and 2.3% for ResNet-152, ViT-B/16, and Mixer-B/16, respectively).

As expected, fewer training examples amplify the drawback of ViTs and MLP-Mixers' lack of the convolutional inductive bias — their accuracies decline much faster than ResNets' (see Figure 4 in the Appendix and the corresponding numbers in Table 5). However, SAM can drastically rescue ViTs and MLP-Mixers' performance decrease on smaller training sets. Figure 4 (right) shows that *the improvement brought by SAM over vanilla SGD training is proportional to the number of training images.* When trained on i1k (1/4), it boosts ViT-B/16 and Mixer-B/16 by 14.4% and 25.6%, escalating their results to 66.8% and 62.8%, respectively. It also tells that ViT-B/16-SAM matches the performance of ResNet-152-SAM even with only 1/2 ImageNet training data.

## 6 CONCLUSIONS AND LIMITATIONS

This paper presents a detailed analysis of the convolution-free ViTs and MLP-Mixers from the lens of the loss landscape geometry, intending to reduce the models' dependency on massive pre-training and/or strong data augmentations. We arrive at the sharpness-aware minimizer (SAM) after observing sharp local minima of the converged models. By explicitly regularizing the loss geometry through SAM, the models enjoy much flatter loss landscapes and improved generalization regarding accuracy and robustness. The resultant ViT models outperform ResNets of comparable size and throughput when learned with no pre-training or strong augmentations. Further investigation reveals that the smoothed loss landscapes attribute to much sparser activated neurons in the first few layers. Last but not least, we discover that SAM and strong augmentations share certain similarities to enhance the generalization. They both smooth the average loss curvature and encourage linearity.

Despite achieving better generalization, training ViTs with SAM has the following limitations which could lead to potential future work. First, SAM incurs another round of forward and backward propagations to update $\epsilon$, which will lead to around 2x computational cost per update. Second, we notice that the effect of SAM diminishes as the training dataset becomes larger, so it is vital to develop learning algorithms that can improve/accelerate the large-scale pre-training process.

## ETHICS STATEMENT

We are not aware of any immediate ethical issues in our work. We hope this paper can provide new insights into the convolution-free neural architectures and their interplay with optimizers, hence benefiting future developments of advanced neural architectures that are efficient in data and computation. Possible negative societal impacts mainly hinge on the applications of convolution-free architectures, whose societal effects may translate to this work.

## ACKNOWLEDGEMENT

This work is partially supported by NSF under IIS-1901527, IIS-2008173, IIS-2048280 and by Army Research Laboratory under agreement number W911NF-20-2-0158.

## REPRODUCIBILITY STATEMENT

We provide comprehensive experimental details and references to existing works and codebases to ensure reproducibility. The specification of all the architectures used in this paper is available in Appendix A. The instructions for plotting the landscape and the attention map are detailed in Appendix E. We also present our approach to approximating Hessian's dominant eigenvalue $\lambda_{max}$ and the NTK condition number in Appendices F and G, respectively. Finally, Appendix H describes all the necessary training configurations, data augmentations, and SAM hyperparameters to ensure the reproducibility of our results.

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

APPENDICES

## A  ARCHITECTURES

Table 8 specifies the ViT (Dosovitskiy et al., 2021; Vaswani et al., 2017) and MLP-Mixer (Tolstikhin et al., 2021) architectures used in this paper. "S" and "B" denote the small and base model scales following (Dosovitskiy et al., 2021; Touvron et al., 2021b; Tolstikhin et al., 2021), followed by the size of each image patch. For instance, "B/16" means the model of base scale with non-overlapping image patches of resolution $16 \times 16$. We use the input resolution $224 \times 224$ throughout the paper. Following Tolstikhin et al. (2021), we sweep the batch sizes in $\{32, 64, \dots, 8192\}$ on TPU-v3 and report the highest throughput for each model.

Table 7: Comparison under the adversarial training framework on ImageNet (numbers in the parentheses denote the improvement over the standard adversarial training without SAM). With similar model size and throughput, ViTs-SAM can still outperform ResNets-SAM for clean accuracy and adversarial robustness.

| Model | #params | Throughput (img/sec/core) | ImageNet | Real | V2 | PGD-10 | ImageNet-R | ImageNet-C |
|---|---|---|---|---|---|---|---|---|
| **ResNet** | | | | | | | | |
| ResNet-50-SAM | 25M | 2161 | 70.1 (-0.7) | 77.9 (-0.3) | 56.6 (-0.8) | 54.1 (+0.9) | 27.0 (+0.9) | 42.7 (-0.1) |
| ResNet-101-SAM | 44M | 1334 | 73.6 (-0.4) | 81.0 (+0.1) | 60.4 (-0.6) | 58.8 (+1.4) | 29.5 (+0.6) | 46.9 (+0.3) |
| ResNet-152-SAM | 60M | 935 | 75.1 (-0.4) | 82.3 (+0.2) | 62.2 (-0.4) | 61.0 (+1.8) | 30.8 (+1.4) | 49.1 (+0.6) |
| **Vision Transformer** | | | | | | | | |
| ViT-S/16-SAM | 22M | 2043 | 73.2 (+1.2) | 80.7 (+1.7) | 60.2 (+1.4) | 58.0 (+5.2) | 28.4 (+2.4) | 47.5 (+1.6) |
| ViT-B/32-SAM | 88M | 2805 | 69.9 (+3.0) | 76.9 (+3.4) | 55.7 (+2.5) | 54.0 (+6.4) | 26.0 (+3.0) | 46.4 (+3.0) |
| ViT-B/16-SAM | 87M | 863 | 76.7 (+3.9) | 82.9 (+4.1) | 63.6 (+4.3) | 62.0 (+7.7) | 30.0 (+4.9) | 51.4 (+5.0) |
| **MLP-Mixer** | | | | | | | | |
| Mixer-S/16-SAM | 18M | 4005 | 67.1 (+2.2) | 74.5 (+2.3) | 52.8 (+2.5) | 50.1 (+4.1) | 22.9 (+2.6) | 37.9 (+2.5) |
| Mixer-B/32-SAM | 60M | 4209 | 69.3 (+9.1) | 76.4 (+10.2) | 54.7 (+9.4) | 54.5 (+13.9) | 26.3 (+8.0) | 43.7 (+8.8) |
| Mixer-B/16-SAM | 59M | 1390 | 73.9 (+11.1) | 80.8 (+11.8) | 60.2 (+11.9) | 59.8 (+17.3) | 29.0 (+10.5) | 45.9 (+12.5) |

Table 8: Specifications of the ViT and MLP-Mixer architectures used in this paper. We train all the architectures with image resolution $224 \times 224$.

| Model | #params | Throughput (img/sec/core) | Patch Resolution | Sequence Length | Hidden Size | #heads | #layers | Token MLP Dimension | Channel MLP Dimension |
|---|---|---|---|---|---|---|---|---|---|
| ViT-S/32 | 23M | 6888 | $32 \times 32$ | 49 | 384 | 6 | 12 | – | – |
| ViT-S/16 | 22M | 2043 | $16 \times 16$ | 196 | 384 | 6 | 12 | – | – |
| ViT-S/14 | 22M | 1234 | $14 \times 14$ | 256 | 384 | 6 | 12 | – | – |
| ViT-S/8 | 22M | 333 | $8 \times 8$ | 784 | 384 | 6 | 12 | – | – |
| ViT-B/32 | 88M | 2805 | $32 \times 32$ | 49 | 768 | 12 | 12 | – | – |
| ViT-B/16 | 87M | 863 | $16 \times 16$ | 196 | 768 | 12 | 12 | – | – |
| Mixer-S/32 | 19M | 11401 | $32 \times 32$ | 49 | 512 | – | 8 | 256 | 2048 |
| Mixer-S/16 | 18M | 4005 | $16 \times 16$ | 196 | 512 | – | 8 | 256 | 2048 |
| Mixer-S/8 | 20M | 1498 | $8 \times 8$ | 784 | 512 | – | 8 | 256 | 2048 |
| Mixer-B/32 | 60M | 4209 | $32 \times 32$ | 49 | 768 | – | 12 | 384 | 3072 |
| Mixer-B/16 | 59M | 1390 | $16 \times 16$ | 196 | 768 | – | 12 | 384 | 3072 |
| Mixer-B/8 | 64M | 466 | $8 \times 8$ | 784 | 768 | – | 12 | 384 | 3072 |

## B  WHEN SAM MEETS ADVERSARIAL TRAINING

Interestingly, SAM and adversarial training are both minimax problems except that SAM's inner maximization is with respect to the network weights, while the latter concerns about the input for

Table 9: Hyperparameters for downstream tasks. All models are fine-tuned with $224 \times 224$ resolution, a batch size of 512, cosine learning rate decay, no weight decay, and grad clipping at global norm 1.

| Dataset | Total steps | Warmup steps | Base LR |
|---|---|---|---|
| CIFAR-10 | 10K | 500 | |
| CIFAR-100 | 10K | 500 | $\{0.001, 0.003, 0.01, 0.03\}$ |
| Flowers | 500 | 100 | |
| Pets | 500 | 100 | |

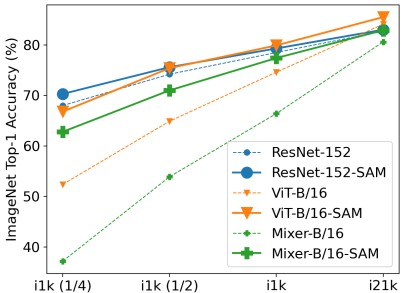 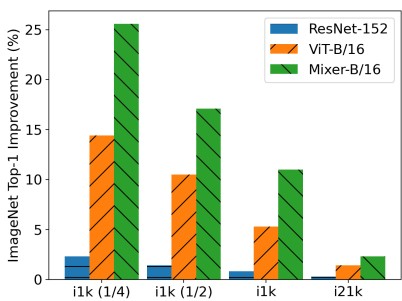

Figure 4: ImageNet accuracy (**Left**) and improvement (**Right**) brought by SAM.

defending contrived attack (Madry et al., 2018; Wong et al., 2020). Moreover, similar to SAM, Shafahi et al. (2019) suggest that adversarial training can flatten and smooth the loss landscape. In light of these connections, we study ViTs and MLP-Mixers under the adversarial training framework (Wu et al., 2020; Madry et al., 2018). We use the fast adversarial training (Wong et al., 2020) (FGSM with random start) with the $l_\infty$ norm and maximum per-pixel change 2/255 during training. All the hyperparameters remain the same as the vanilla supervised training. When evaluating the adversarial robustness, we use the PGD attack (Madry et al., 2018) with the same maximum per-pixel change 2/255. The total number of attack steps is 10, and the step size is 0.25/255. To incorporate SAM, we formulate a three-level objective:

$$\min_w \max_{\epsilon \in \mathbb{S}_{sam}} \max_{\delta \in \mathbb{S}_{adv}} L_{train}(w + \epsilon, x + \delta, y), \tag{5}$$

where $\mathbb{S}_{sam}$ and $\mathbb{S}_{adv}$ denote the allowed perturbation norm balls for the model parameter $w$ and input image $x$, respectively. Note that we can simultaneously obtain the gradients for computing $\epsilon$ and $\delta$ by backpropagation only once. To lower the training cost, we use fast adversarial training (Wong et al., 2020) with the $l_\infty$ norm for $\delta$, and the maximum per-pixel change is set as 2/255.

Table 7 (see Appendices) evaluates the models' clean accuracy, real-world robustness, and adversarial robustness (under 10-step PGD attack (Madry et al., 2018)). It is clear that the landscape smoothing significantly improves the convolution-free architectures for both clean and adversarial accuracy. However, we observe a slight accuracy decrease on clean images for ResNets despite gain for robustness. Similar to our previous observations, *ViTs surpass similar-size ResNets when adversarially trained on ImageNet with Inception-style preprocessing for both clean accuracy and adversarial robustness.*

## C WHEN SAM MEETS CONTRASTIVE LEARNING

In addition to data augmentations and large-scale pre-training, another notable way of improving a neural model's generalization is (supervised) contrastive learning (Chen et al., 2020; He et al., 2020; Caron et al., 2021; Khosla et al., 2020). We couple SAM with the supervised contrastive learning (Khosla et al., 2020) for 350 epochs, followed by fine-tuning the classification head by 90 epochs for both ViT-S/16 and ViT-B/16. We train ViTs under the supervised contrastive learning framework (Khosla et al., 2020). We take the classification token output from the last layer as the encoded representation and retain the structures of the projection and classification heads (Khosla et al., 2020). We employ a batch size 2048 without memory bank (He et al., 2020) and use AutoAugment (Cubuk et al., 2019) with strength 1.0 following Khosla et al. (2020). For the 350-epoch pretraining stage, the contrastive loss temperature is set as 0.1, and we use the LAMB optimizer (You et al., 2020) with learning rate $0.001 \times \frac{\text{batch size}}{256}$ along with a cosine decay schedule. For the second stage, we train the classification head for 90 epochs via a RMSProp optimizer (Tieleman & Hinton, 2012) with base learning rate 0.05 and exponential decay. The weight decays are set as 0.3 and 1e-6 for the first and second stages, respectively. We use a small SAM perturbation strength $\rho = 0.02$.

Compared to the training procedure without SAM, we find considerable performance gain thanks to SAM's smoothing of the contrastive loss geometry, improving the ImageNet top-1 accuracy of ViT-S/16 from 77.0% to 78.1%, and ViT-B/16 from 77.4% to 80.0%. In comparison, the improvement on ResNet-152 is less significant (from 79.7% to 80.0% after using SAM).

Table 10: Accuracy on downstream tasks of the models pre-trained on ImageNet. SAM improves ViTs and MLP-Mixers' transferabilities. ViTs transfer better than ResNets of similar sizes.

| % | ResNet-50-SAM | ResNet-152-SAM | ViT-S/16 | ViT-S/16-SAM | ViT-B/16 | ViT-B/16-SAM | Mixer-S/16 | Mixer-S/16-SAM | Mixer-B/16 | Mixer-B/16-SAM |
|---|---|---|---|---|---|---|---|---|---|---|
| **CIFAR-10** | 97.4 | 98.2 | 97.6 | 98.2 | 98.1 | 98.6 | 94.1 | 96.1 | 95.4 | 97.8 |
| **CIFAR-100** | 85.2 | 87.8 | 85.7 | 87.6 | 87.6 | 89.1 | 77.9 | 82.4 | 80.0 | 86.4 |
| **Flowers** | 90.0 | 91.1 | 86.4 | 91.5 | 88.5 | 91.8 | 83.3 | 87.9 | 82.8 | 90.0 |
| **Pets** | 91.6 | 93.3 | 90.4 | 92.9 | 91.9 | 93.1 | 86.1 | 88.7 | 86.1 | 92.5 |
| **Average** | 91.1 | 92.6 | 90.0 | 92.6 | 91.5 | 93.2 | 85.4 | 88.8 | 86.1 | 91.7 |

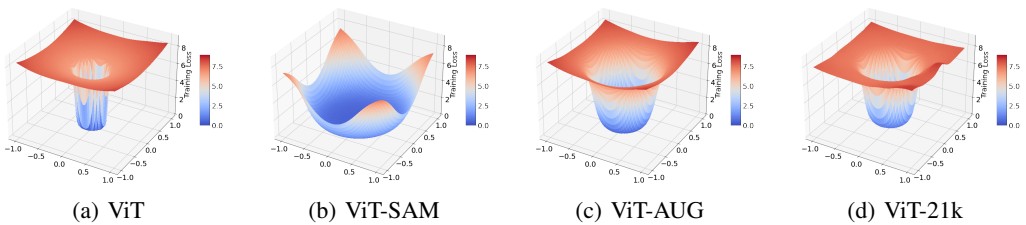

(a) ViT  (b) ViT-SAM  (c) ViT-AUG  (d) ViT-21k

Figure 5: Cross-entropy loss landscapes of ViT-B/16, ViT-B/16-SAM, ViT-B/16-AUG, and ViT-B/16-21k. Strong augmentations and large-scale pre-training can also smooth the curvature.

## D WHEN SAM MEETS TRANSFER LEARNING

We also study the role of smoothed loss geometry in transfer learning. We select four datasets to test ViTs and MLP-Mixers' transferabilities: CIFAR-10/100 (Krizhevsky, 2009), Oxford-IIIT Pets (Parkhi et al., 2012), and Oxford Flowers-102 (Nilsback & Zisserman, 2008). We use image resolution $224 \times 224$ during fine-tuning on downstream tasks, other settings exactly follow Dosovitskiy et al. (2021); Tolstikhin et al. (2021) (see Table 9). Note that we do not employ SAM during fine-tuning. We perform a grid search over the base learning rates on small sub-splits of the training sets (10% for Flowers and Pets, 2% for CIFAR-10/100). After that, we fine-tune on the entire training sets and report the results on the respective test sets. For comparison, we also include ResNet-50-SAM and ResNet-152-SAM in the experiments. Table 10 summarizes the results, which confirm that the enhanced models also perform better after fine-tuning and that MLP-Mixers gain the most from the sharpness-aware optimization.

## E VISUALIZATION

### E.1 LOSS LANDSCAPE

We use the "filter normalization" method (Li et al., 2018) to visualize the loss function curvature in Figure 1 and 5. For a fair comparison, we use the cross-entropy loss when plotting the landscapes for all architectures, although the original training objective is the sigmoid loss for ViTs and MLP-Mixers. Note that their sigmoid loss geometry is even sharper. We equally sample 2,500 points on the 2D projection space and compute the losses using 10% of the ImageNet training images (Chen et al., 2020), i.e., the i1k (1/10) subset in the main text to save computation.

### E.2 ATTENTION MAP

The visualization of the ViT's attention maps (Figure 3 in the main text) follows (Caron et al., 2021). We average the self-attention scores of the "classification token" from the last MSA layer to obtain a matrix $A \in \mathbb{R}^{H/P \times W/P}$, where $H$, $W$, $P$ are the image height, width, and the patch resolution, respectively. Then we upsample $A$ to the image shape $H \times W$ before generating the figure.

Table 11: The SAM perturbation strength $\rho$ for training on ImageNet. ViTs and MLP-Mixers favor larger $\rho$ than ResNets does. Larger models with longer patch sequences need stronger strengths.

| Model | Task | SAM $\rho$ |
|---|---|---|
| ResNet | | |
| ResNet-50-SAM | supervised | 0.02 |
| ResNet-101-SAM | supervised | 0.05 |
| ResNet-152-SAM | supervised | 0.02 |
| ResNet-50x2-SAM | supervised | 0.05 |
| ResNet-101x2-SAM | supervised | 0.05 |
| ResNet-152x2-SAM | supervised | 0.05 |
| ResNet-50-SAM | adversarial | 0.05 |
| ResNet-101-SAM | adversarial | 0.05 |
| ResNet-152-SAM | adversarial | 0.05 |
| ViT | | |
| ViT-S/32-SAM | supervised | 0.05 |
| ViT-S/16-SAM | supervised | 0.1 |
| ViT-S/14-SAM | supervised | 0.1 |
| ViT-S/8-SAM | supervised | 0.15 |
| ViT-B/32-SAM | supervised | 0.15 |
| ViT-B/16-SAM | supervised | 0.2 |
| ViT-B/16-AUG-SAM | supervised | 0.05 |
| ViT-S/16-SAM | adversarial | 0.1 |
| ViT-B/32-SAM | adversarial | 0.1 |
| ViT-B/16-SAM | adversarial | 0.1 |
| ViT-S/16-SAM | supervised contrastive | 0.02 |
| ViT-B/16-SAM | supervised contrastive | 0.02 |
| MLP-Mixer | | |
| Mixer-S/32-SAM | supervised | 0.1 |
| Mixer-S/16-SAM | supervised | 0.15 |
| Mixer-S/8-SAM | supervised | 0.2 |
| Mixer-B/32-SAM | supervised | 0.35 |
| Mixer-B/16-SAM | supervised | 0.6 |
| Mixer-B/8-SAM | supervised | 0.6 |
| Mixer-B/16-AUG-SAM | supervised | 0.2 |
| Mixer-S/16-SAM | adversarial | 0.05 |
| Mixer-B/32-SAM | adversarial | 0.25 |
| Mixer-B/16-SAM | adversarial | 0.25 |

## F  HESSIAN EIGENVALUE

The Hessian matrix requires second-order derivative, so we compute the Hessian (and all the sub-diagonal Hessian) $\lambda_{max}$ using 10% of the ImageNet training images (i.e., i1k (1/10)) via power iteration [1], where we use 100 iterations to ensure its convergence.

## G  NTK CONDITION NUMBER

We approximate the neural tangent kernel on the i1k (1/10) subset by averaging over block diagonal entries (with block size $48 \times 48$) in the full NTK. Notice that the computation is based on the architecture at initialization without training. As the activation plays an important role when computing NTK — we find that smoother activation functions enjoy smaller condition numbers, we replace the GELU in ViT and MLP-Mixer with ReLU for a fair comparison with ResNet.

## H  TRAINING DETAILS

We use image resolution $224 \times 224$ during fine-tuning on downstream tasks, other settings exactly follow (Dosovitskiy et al., 2021; Tolstikhin et al., 2021) (see Table 9). Note that we do not employ SAM during fine-tuning. We perform a grid search over the base learning rates on small sub-splits of the training sets (10% for Flowers and Pets, 2% for CIFAR-10/100). After that, we fine-tune on the entire training sets and report the results on the respective test sets.

---

[1] https://en.wikipedia.org/wiki/Power_iteration

Table 12: Hyperparameters for training from scratch on ImageNet with basic Inception-style pre-processing and $224 \times 224$ image resolution.

| | ResNet | ViT | MLP-Mixer |
|---|---|---|---|
| Data augmentation | | Inception-style | |
| Input resolution | | $224 \times 224$ | |
| Batch size | | 4,096 | |
| Epoch | 90 | 300 | 300 |
| Warmup steps | 5K | 10K | 10K |
| Peak learning rate | $0.1 \times \frac{\text{batch size}}{256}$ | 3e-3 | 3e-3 |
| Learning rate decay | cosine | cosine | linear |
| Optimizer | SGD | AdamW | AdamW |
| SGD Momentum | 0.9 | – | – |
| Adam $(\beta_1, \beta_2)$ | – | (0.9, 0.999) | (0.9, 0.999) |
| Weight decay | 1e-3 | 0.3 | 0.3 |
| Dropout rate | 0.0 | 0.1 | 0.0 |
| Stochastic depth | – | – | 0.1 |
| Gradient clipping | – | 1.0 | 1.0 |

Table 13: ImageNet top-1 accuracy (%) of ViT-B/16 and Mixer-B/16 when trained from scratch with different perturbation strength $\rho$ in SAM.

| SAM $\rho$ | 0.0 | 0.05 | 0.1 | 0.2 | 0.25 | 0.35 | 0.4 | 0.5 | 0.6 | 0.65 |
|---|---|---|---|---|---|---|---|---|---|---|
| ViT-B/16 | 74.6 | 77.5 | 78.8 | **79.9** | 79.3 | – | – | – | – | – |
| Mixer-B/16 | 66.4 | 69.5 | – | – | 74.1 | 74.7 | 75.6 | 76.9 | **77.4** | 77.1 |

Except for the experiments in Section 4.5 (SAM with strong data augmentations) and Appendix C (contrastive learning), we train all the models from scratch on ImageNet with the basic Inception-style preprocessing (Szegedy et al., 2016), i.e., a random image crop and a horizontal flip with probability 50%. Please see Table 12 for the detailed training settings. We simply follow the original training settings of ResNet and ViT (Kolesnikov et al., 2020; Dosovitskiy et al., 2021). For MLP-Mixer, we remove the strong augmentations in its original training pipeline and perform a grid search over the learning rate in $\{0.003, 0.001\}$, weight decay in $\{0.3, 0.1, 0.03\}$, Dropout rate in $\{0.1, 0.0\}$, and stochastic depth in $\{0.1, 0.0\}$. Note that training for 90 epochs is enough for ResNets to converge, and longer schedule brings almost no effect. For all the experiments, we use 128 TPU-v3 cores (2 per chip), resulting in 32 images per core. The SAM computation for $\hat{\epsilon}$ is conducted on each core independently.

## H.1 PERTURBATION STRENGTH IN SAM

Different architecture species favor different strengths of perturbation $\rho$. We perform a grid search over $\rho$ and report the best results — Table 11 reports the corresponding strengths used in our ImageNet experiments. Besides, we show the results when varying $\rho$ in Table 13. Similar to (Foret et al., 2021), we also find that a relative small $\rho \in [0.02, 0.05]$ works the best for ResNets. However, larger $\rho$ gives rise to the best results for ViTs and MLP-Mixers. We also observe that architectures with larger capacities and longer input sequences prefer stronger perturbation strengths. Interestingly, the choice of $\rho$ coincides with our previous observations. Since MLP-Mixers suffer the sharpest landscapes, they need the largest perturbation strength. As strong augmentations and contrastive learning already improve generalization, the suitable $\rho$ becomes significantly smaller. Note that we do not re-tune any other hyperparameters when using SAM.

## H.2 TRAINING ON IMAGENET SUBSETS

In Section 5.1, we train the models on ImageNet subsets, and the hyperparameters have to be adjusted accordingly. We simply change the batch size to maintain similar total iterations and keep all other settings the same, i.e., 2048 for i1k (1/2), 1024 for i1k (1/4), and 512 for i1k (1/10). We do not scale the learning rate as we find the scaling harms the performance.

### H.3 TRAINING WITH STRONG AUGMENTATIONS

We tune the learning rate and regularization when using strong augmentations (mixup with probability 0.5, RandAugment with two layers and magnitude 15) in Section 4.5 following (Tolstikhin et al., 2021). For ViT, we use 1e-3 peak learning rate, 0.1 weight decay, 0.1 Dropout, and 0.1 stochastic depth; For MLP-Mixer, those hyperparameters are exactly the same as (Tolstikhin et al., 2021), peak learning rate as 1e-3, weight decay as 0.1, Dropout as 0.0, and stochastic depth as 0.1. Other settings are unchanged (Table 12).

## I LONGER SCHEDULE OF VANILLA SGD

Since SAM needs another forward and backward propagation to compute $\hat{\epsilon}$, its training overhead is $\sim 2\times$ of the vanilla baseline. We also experiment with $2\times$ schedule vanilla training (600 epochs). We observe that training longer brings no effect on both clean accuracy and robustness, indicating that the current 300 training epochs for ViTs and MLP-Mixers are enough for them to converge.

## J VARYING WEIGHT DECAY STRANGTH

Table 14: ImageNet accuracy and curvature analysis for ViT-B/16 when we vary the weight decay strength in Adam (AdamW).

| Model | Weight decay | ImageNet (%) | $\|w\|_2$ | $L_{train}$ | $L_{train}^{\mathcal{N}}$ | $\lambda_{max}$ |
|---|---|---|---|---|---|---|
| ViT-B/16 | 0.2 | 74.2 | 339.8 | 0.51 | 4.22 | 507.4 |
| | 0.3 | 74.6 | 269.3 | 0.65 | 6.66 | 738.8 |
| | 0.4 | 74.7 | 236.7 | 0.77 | 7.08 | 1548.9 |
| | 0.5 | 74.4 | 211.8 | 0.98 | 7.21 | 2251.7 |
| ViT-B/16-SAM | 0.2 | 79.9 | 461.4 | 0.69 | 0.72 | 13.1 |
| | 0.3 | 79.9 | 353.8 | 0.82 | 0.96 | 20.9 |
| | 0.4 | 79.4 | 301.1 | 0.85 | 0.98 | 26.1 |
| | 0.5 | 78.7 | 259.6 | 0.95 | 1.33 | 45.5 |

In this section, we vary the strength of weight decay and see the effects of this commonly used regularization approach. As shown in Table 14, weight decay helps improve the accuracy on ImageNet when training without SAM, the weight norm also decreases when we enlarge the decay strength as expected. However, enlarging the weight decay aggravates the problem of converging to a sharper region measured by both $L_{train}^{\mathcal{N}}$ and $\lambda_{max}$. Another observation is that $\|w\|_2$ consistently increases after applying SAM for every weight decay strength in Table 14, together with the improved ImageNet accuracy and smoother landscape curvature.

