# OpenReview forum: "When Vision Transformers Outperform ResNets without Pre-training or Strong Data Augmentations"
_ICLR.cc/2022/Conference — ICLR 2022 Spotlight_

### Official Review · Reviewer_y3Uf · 2021-10-21

**Correctness:** 4
**Technical Novelty And Significance:** 2
**Empirical Novelty And Significance:** 3
**Recommendation:** 5
**Confidence:** 4

**Main Review:**

+ The paper has conducted an abundance of experiments on different tasks to demonstrate the effectiveness of the sharpness-aware minimization (SAM).
+ The authors present a comprehensive analysis on SAM. The differences and similarities between SAM and data augmentation are well explained.

- This paper uses the existing idea of sharpness-aware minimization to alleviate the dependency of massive pre-training and strong augmentation, which makes the novelty limited.



**Summary Of The Paper:**

This paper alleviate the dependency on massive pre-training and data augmentation. It promotes the smoothness using a recently proposed sharpness-aware optimizer to improve the accuracy and robustness of the ViTs and MLP-Mixers. This paper has demonstrated that the sharpness-aware optimizer could be leveraged to boost the performance of ViTs and MLP-Mixers without pretraining and strong augmentaitons on different tasks including supervised, adversarial, contrastive, and transfer learning.

**Summary Of The Review:**

Overall, this paper is well writen. The overview of SAM and the explanation of some observations are clearly presented. Besides, the authors demonstrate the effectivenss of SAM in various tasks by showing a large amount of experimental results. The experimental part is comprehensive and convincing. The major drawback of this paper is that the SAM is an existing idea, even though the authors use it to overcome the issue brought by heavy pre-training and data augmentations. Therefore, this paper is like proving the effectiveness of an existing idea by doing many experimental validations.

---

> ### Author Response · Authors · 2021-11-16
> **Responses to Reviewer y3Uf**
>
> Thanks for your constructive feedback, below are our responses.
>
> Previously, the training of ViTs and Mixers required excessive data, computing, and sophisticated tuning of many hyperparameters for augmentations. Through comprehensive analyses, we show that convolution-free architectures suffer poor trainability and converge to extreme sharp regions (for both average and worst-case curvature) that usually cannot generalize well. This finding is nontrivial, along with the battery of tools for analyses --- as summarized by Reviewer hGsf: loss landscape visualization, number of parameters, neural tangent kernel (NTK), Hessian matrix, training dynamics, percentage of active neurons, throughput, norm of weights and activations, attention map visualizations and missing rate under linear interpolation.
>
> Besides identifying SAM that can greatly enhance performance, we study the intrinsic property changes after SAM including weight norm, attention map, and model sparsity, which are meaningful and insightful for future development of learning algorithms.
>
> Finally, we study how SAM interplay with strong augmentations, scale of training datasets, contrastive learning, etc. The results provide empirical foundations for future development of training recipes, e.g., how to balance optimizers and data, regularization, and augmentations.

---

> ### Author Response · Authors · 2021-12-03
> **Follow-up questions**
>
> Dear reviewer y3Uf,
>
> We added more materials in the revised paper and made it more rigorous.
> We are just wondering are there any updates or follow-up questions about our paper?
> Thanks again for your detailed and constructive reviews.

---

### Official Review · Reviewer_CmdC · 2021-11-01

**Correctness:** 3
**Technical Novelty And Significance:** 3
**Empirical Novelty And Significance:** 3
**Recommendation:** 8
**Confidence:** 5

**Main Review:**

This paper is well written and organized.  The motivation comes from the loss landscapes comparison for different network architecture such as ResNet, ViT and Mixer.  The visualization has revealed the problem of the current ViT and Mixer as both of them tend to converge at sharp minima which will limit the generalization capabilities.  As the issue has been clearly pointed out, the solution is straightforward: avoid the sharp local minima when training ViT.  Such that this paper adopts SAM as an answer.  Although, SAM is from existing work, this paper gives a good explanation on why among different optimizer, SAM could be a good choice.

In addition to the method, this paper has shown that on image classification,  contrastive learning and adversarial training ViT can be consistently improved with SAM.  Especially, the visualization of attention map w/ SAM, the ViT feature map does contain more semantics.

Concerns: I agree with that current ViT tends to converge at sharp local minima.  It could be the problem from the usage of existing optimizer.  On the other hand, the problem may also comes from the nature of current ViT design.  Recent vision transformers such as SWIN, PVTv2, VOLO, all of them has clearly shown that by optimizing the architecture of ViT, it can also avoid the local minima.  Based on current manuscript , it is unknown that how SAM can be used for SWIN or PVTv2 and whether SAM can be used as a general optimizer for arbitrary vision transformer training.

[1] Swin Transformer: Hierarchical Vision Transformer using Shifted Windows

[2] PVTv2: Improved Baselines with Pyramid Vision Transformer

[3] VOLO: Vision Outlooker for Visual Recognition

**Summary Of The Paper:**

As the success of Vision Transformer ViT has shown its potentials for computer vision tasks, this paper investigates a more effective way of training a ViT understand a standard ImageNet pre-training setting such as no extra training data and no strong data augmentation.  In general, without those conditions, a typical ViT can not perform as good as widely convolutional based network architectures such as ResNet.  This paper addresses the issue from the point of loss landscape and then propose to use a better optimization strategy named Sharp-aware minimizer (SAM) for ViT related architecture optimization.  With the proposed SAM, ViT can achieve better accuracy significantly understand standard ImageNet training/testing protocol.  In addition to the improved performance on ImageNet, this paper further shows the visualization of the attention map and the improved performance on other applications such as contrastive learning and adversarial training.

**Summary Of The Review:**

In overall, I vote for acceptance for this paper but I think it can be further improved.
In the experiments, ResNet-SAM results are promising, the claim could be stronger if other vision transformers can be evaluated as ViT has been widely argued for its inefficient architecture design.

---

> ### Author Response · Authors · 2021-11-16
> **Responses to Reviewer Cmdc**
>
> Thanks for your constructive and positive feedback, below are our responses.
>
> - **It could be the problem from the usage of the existing optimizer.**
> The superiority of the adaptive learning algorithm over (momentum) SGD when training attention models has been shown before [1, 2]. We also try to train a ViT-B/16 with momentum SGD. However, it only achieves 71.5\% accuracy on ImageNet (although we perform extensive hyperparameter tuning for the learning rate and the $l_2$ normalization), which is much worse than Adam (74.6\%). We surprisingly find that SGD + SAM achieves 79.1\% accuracy on ImageNet, making the gap between SGD and Adam largely reduced. The experiment indicates that SAM can enable ViTs to be effectively trained with SGD. We include the results in the revision.
>
> - **Recent vision transformers such as SWIN, PVTv2, VOLO, all of them have clearly shown that by optimizing the architecture of ViT, it can also avoid the local minima.**
> The training pipelines for Swin, PVTv2, and VOLO all have strong augmentations (e.g., RangAugment), which can also smooth the average loss geometry according to our study (see Table 6). Hence, it is not clear whether the modification on the architecture or the augmentation alleviated the problem of converging to sharp regions. Furthermore, we show in Table 5 that combining SAM and strong augmentations can lead to larger improvements on ImageNet for ViT-B/16 and Mixer-B/16. In our revision, we also demonstrate that SAM can still be beneficial after introducing convolution to ViT (see Table 3): a R50-B/16 achieves 82.4\% accuracy when trained from scratch on ImageNet without any strong augmentations.
>
> [1] Zhang et al. Why are Adaptive Methods Good for Attention Models? NeurIPS 2020.
> [2] Liu et al. Understanding the Difficulty of Training Transformers. EMNLP 2020.

---

### Official Review · Reviewer_1uek · 2021-11-03

**Correctness:** 3
**Technical Novelty And Significance:** 2
**Empirical Novelty And Significance:** 4
**Recommendation:** 8
**Confidence:** 4

**Main Review:**

**Update:** I have increased my score from a 3 to an 8 after the the authors' response

SAM is a new and promising method, so it’s important to examine and understand its effects. And the benefits of SAM for ViTs and MLP-Mixers appear significant. I applaud the breadth of the authors’ attempts: they examine the effects of SAM across a wide range of image classification scenarios and using a variety of analysis methods.

I am convinced that SAM works well for ViTs and MLP-Mixers. But I’m not convinced of why it works. My main concern about this paper is that it is insufficiently rigorous. The authors provide a number of explanations that are not justified by the evidence, and none of the experiments are run with replicates, making it difficult to determine whether observed differences are meaningful. I am not confident that this work is suitable for publication in its current state. However, I am optimistic that it could be suitable for publication after revision.

# Detailed Feedback

## Replicates and measures of variability

Ideally, all the experiments should be run in replicate with 10 different seeds. However, I understand that this is computationally costly, so I would accept running replicates (three at a bare minimum) for the six core model configurations: ViT-B/16, MLP-Mixer-B/16, and ResNet-152 with and without SAM. Results with these models should be accompanied by measures of variability (or simply listing all three numbers, if only 3 replicates are used).

## Unjustified claims

There are numerous unjustified scientific claims made, which I detail as follows:

End of first paragraph in **Section 3**:
>“There also exists a large gap between ViTs and ResNets in robustness tests.

Requires a reference.

In **Section 3. Small training errors** the authors claim that

>using the cross-token MLP to learn the interplay across image patches is more prone to overfitting than ViTs’ self-attention mechanism whose behavior is restricted by a softmax.

This claim seems speculative, and should be tested. The authors could train a ViT with a different attention normalization scheme, or without one altogether. Learning Q, K, and V projections as opposed to learning MLP. At the very least, a convincing theoretical explanation should be provided.

In **Section 4.2. Higher Accuracy**, the authors state:
>Empirically, the degree of improvement negatively correlates with the level of inductive biases built into the architecture. ResNets with inherent translation equivalence and locality benefit less from landscape smoothing than the attention-based ViTs.

This is an interesting claim, but there are multiple issues with it. First, “level of inductive bias” is vague. I would suggest the language be changed to “correlates with expressivity” or “negatively correlates with the constraints of the inductive biases“. The “level” or “severity” of an inductive bias can be orthogonal to the constraints it imposes on expressivity. ConViT (d’Ascoli et al., 2021), for example, has a convolutional inductive bias but is completely unconstrained in that it can learn to be a vanilla ViT. Second, this claim can easily be tested. The authors could examine the effects of SAM when varying model architecture choices. For example, exchanging layer norm for batch norm in ResNets would have little effect on the model’s expressivity, and consequently the authors’ hypothesis predicts that the effect of SAM should not vary significantly. The authors could examine the effect of SAM on other architectures with varying (and controllable) inductive biases, such as ConViT (d’Ascoli et al., 2021), CvT (Wu et al., 2021), CMT (Guo et al., 2021), a ViT with a convolutional stem (Xiao et al., *Early Convolutions Help Transformers See Better*, 2021), or a CNN-reparameterized into an MLP á la (d’Ascoli et al., *Finding the Needle in the Haystack with Convolutions*, 2020).

The effects of SAM on activation norm (Table 3) in the ViT are not consistent, so I think the authors should avoid making the claim that “we find that the norm of the post-activation value...become even bigger” (*Section 4.4 Greater Weight Norms*).

The authors state

>Interestingly, the ViT model optimized with SAM can encode plausible segmentation information, giving rise to better interpretability than the one trained via the conventional SGD optimization

Claims regarding segmentation should be evaluated with a segmentation task (e.g. results using a transformer backbone for segmentation with vs. without SAM), and claims about interpretability should be evaluated with controlled human experiments (see Leavitt and Morcos, *Towards Falsifiable Interpretability*). I think the strongest claim that could be made without appropriate experiments is “it appears more interpretable”.

Table 11 shows that the ViT model is trained using Adam, not SGD, but you write "...giving rise to better interpretability than the one trained via the conventional SGD optimization".

Transformers are notoriously unstable during training. ADAM is thought to compensate for this instability. It would be interesting to see whether SAM allows vision transformers to be trained with SGD.

Does Figure 2a depict training with or without SAM? I think it would be informative to plot both to depict the effect of SAM over the course of training.

The following claims are made in the introduction, some of which I address in my above comments:

>We conjecture that the convolution-induced translation equivariance and locality help ResNets escape from bad local minima when trained on visual data.

This claim is not justified: the effects of translation equivariance and locality on SAM are never tested directly, nor is the concept of a “bad” local minimum clearly defined. As suggested before, the authors should repeat their analyses using vision transformers with convolutional/local inductive biases. The authors should also clearly define what they mean by a “bad” local minimum.

>By analyzing some intrinsic model properties, we find that the models after SAM reduce the Hessian eigenvalues by activating sparser neurons (on ImageNet), especially in the first few layers.

This language implies a causal chain that SAM increases sparsity, and the increased sparsity reduces Hessian eigenvalues. The current experiments can only justify the claim that “SAM increases sparsity and reduces Hessian eigenvalues”. The authors need to conduct experiments showing that SAM causes sparsity and sparsity causes a reduction in hessian eigenvalues. This could be done by regularizing to increase/decrease sparsity with vs. without SAM, and examining the effects on Hessian eigenvalues and accuracy. Otherwise, the authors should amend this claim to remove the implied SAM->sparsity->Hessian causality.

>The weight norms increase, implying the commonly used weight decay may not be an effective regularization alone.

Changes in weight norms alone are insufficient to justify this claim. The authors should repeat their analyses with varying amounts of weight decay with vs. without SAM.

>A side observation is that, unlike ResNets and MLP-Mixers, ViTs have extremely sparse active neurons, revealing the redundancy of input image patches and the capacity for network pruning.

It’s not clear to me how sparsity translates to image patch redundancy. This is a very interesting idea, and should be better explained.

>Another interesting finding is that ViTs’ performance gain also translates to plausible attention maps containing more perspicuous information about semantic segmentation.

I address this in an earlier comment: evaluate this claim with a segmentation task and/or human interpretability experiments.

## Other Feedback

Analyzing a vision transformer with an inductive bias for locality such as ConViT (d’Ascoli et al., 2021), CvT (Wu et al., 2021), and/or CMT (Guo et al., 2021) could strengthen a lot of the claims in the paper and bridge the results from ViTs to ResNets.

The authors state that

>These results show that both SAM and augmentations make the loss landscape flat on average. The difference is that SAM enforces the smoothness by reducing the largest curvature via a minimax formulation to optimize the worst-case scenario, while augmentations ignore the worse-case curvature and instead smooth the landscape over the directions concerning the inductive biases induced by the augmentations.

This distinction between average- and worst-case curvature is an interesting result. I think the authors should extend their analysis and assess the average flatness of other models with vs. w/o SAM.

In *Section 4.2. Better Robustness*, the effects of SAM on robustness in ResNets should be mentioned. Furthermore, percent changes in evaluation metrics such as accuracy must be specified as relative or absolute (i.e. percentage point). Finally, the authors need to control for the effect of the clean accuracy improvement: is the change in ImageNet-C accuracy larger than expected given the baseline change in ImageNet accuracy?

All the plots in Figure 2 should also contain a ResNet-152, with and without SAM.

I understand that there are space constraints, but the ViT sparsity results should be depicted in a figure. Perhaps consider combining Figs. 2c&d to make more space.

The presentation of the contrastive learning results (Section 5.2) should include results for the baseline, ResNet-152.

Figure 4a should show values both with and without SAM

It’s not clear to me how the sparsity of Mixer activations “suggests the potential redundancy
of image patches.”

The authors should include a discussion of limitations of their work.


**Summary Of The Paper:**

The authors analyze the effects of sharpness-aware minimization (SAM) when applied to vision transformers (ViTs) and MLP-Mixers. They find that the converged loss surface of ViTs and MLP-Mixers is sharp compared to ResNets, and that SAM ameliorates this issue, yielding improved validation accuracy on ImageNet. They then show that SAM improves the performance of ViTs and MLP-Mixers in a variety of image classification scenarios including adversarial attacks, naturalistic corruptions, contrastive training, and transfer learning. They also examine the effect of SAM on activation sparsity, activation maps, and the relationships between different model architecture components and loss surface sharpness.

**Summary Of The Review:**

MLP-Mixers and ViTs are more difficult to train than CNNs. This study shows that SAM is very effective for training MLP-Mixers and ViTs, and attempts to understand why. These could all be important contributions to computer vision. I am convinced that SAM works well for ViTs and MLP-Mixers. But this paper is insufficiently rigorous, leaving me unconvinced of why it works. I don't think its suitable for publication in its current state, but it certainly could be after revisions.

---

> ### Author Response · Authors · 2021-11-16
> **Responses to Reviewer 1uek**
>
> Thank you for your constructive feedback. Many of the suggested experiments are actually helpful for supporting our claims.
>
> - **Replicates and measures of variability**
> We run the experiments for ResNet-152, ViT-B/16, and Mixer-B/16 with and without SAM using three random seeds and report the mean and standard deviation below:
>
> | ResNet-152     	| ResNet-152-SAM 	| ViT-B/16       	| ViT-B/16-SAM  	| Mixer-B/16   	| Mixer-B/16-SAM    	|
> |----------------	|----------------	|----------------	|---------------	|--------------	|---------------	|
> | $78.5 \pm 0.2$ 	| $79.2 \pm 0.2$ 	| $74.5 \pm 0.3$ 	| $79.9\pm 0.2$ 	| $66.6\pm 0.4$ 	| $77.5\pm 0.3$ 	|
>
> The variance is comparably small considering the improvement brought by SAM.
>
> In the following, we provide justifications and modifications for our claims in the paper to make them more rigorous.
> - **“There also exists a large gap between ViTs and ResNets in robustness tests.”**
> We refer to our results for ImageNet-R and ImageNet-C in Table 2. For example, the accuracy of ViT-B/16 (46.6%) lays far behind that of ResNet-152 (50.0%) on ImageNet-C, which is widely used to test a model’s real-world robustness. We add the reference in the revised paper.
>
> - **“using the cross-token MLP to learn the interplay across image patches is more prone to overfitting than ViTs’ self-attention mechanism whose behavior is restricted by a softmax.”**
> As shown in Figure 2 (left), Mixers achieve lower training losses but higher test errors than ViTs, indicating that Mixers are more prone to overfitting than ViTs empirically. To test whether this is caused by the softmax operation, we further train a variation to ViT-B/16 by simply removing the softmax, and we name this model ViT-B/16-Free. The obtained model has lower training error compared with the original ViT-B/16 (0.56 vs. 0.65) but achieves much lower test accuracy on ImageNet (70.5\% vs. 74.6\%).
> Its $L_{train}^\mathcal{N}$ and $\lambda_{max}$ are 7.01 and 1236.2 respectively, revealing that this new model converges to a sharper region than ViT-B/16 ($L_{train}^\mathcal{N}$ is 6.66 and $\lambda_{max}$ is 738.8) both on average and in the worst-case direction. The experiment indicates that the softmax restriction helps alleviate overfitting and the convergence to a sharp region. We add this new experiment in the revised paper.
>
> - **“Empirically, the degree of improvement negatively correlates with the level of inductive biases built into the architecture. ResNets with inherent translation equivalence and locality benefit less from landscape smoothing than the attention-based ViTs.”**
> We agree that “the constraints of the inductive biases” is more rigorous and we change the main text accordingly. To further validate this argument, we use SAM to train the hybrid architecture where the Transformer takes the feature maps of a CNN as the input (see Table 3 of the revised paper). We observe that the improvement brought by SAM decreases after we introduce the convolution to ViT. For example, the accuracy improvement on ImageNet is +2.7\% for R50-ViT/16, which is smaller than that for ViT-B/16 (+5.3\%) but larger than that for ResNet-50 (+0.7\%) and ResNet-152 (+0.8%).
>
> - **“we find that the norm of the post-activation value...become even bigger”**
> In Table 4, only the norm of $a_{12}$ in ViT-B/16 enlarges after using SAM. So we add “In most cases” before this sentence in the main text.
>
> - **“Interestingly, the ViT model optimized with SAM can encode plausible segmentation information, giving rise to better interpretability than the one trained via the conventional SGD optimization”**
> We agree that this sentence is overly strong, so we modify it to “it appears more interpretable” per your advice. We also modify the word SGD to Adam. Due to the time constraint, we are not able to conduct a well-controlled human experiment.
>
> - **“We conjecture that the convolution-induced translation equivariance and locality help ResNets escape from bad local minima when trained on visual data.”**
> We remove this claim in the paper as we discover that hybrid architectures (ResNet+ViT) can also converge to sharp regions. We leave this as an interesting future direction.
>
> - **“By analyzing some intrinsic model properties, we find that the models after SAM reduce the Hessian eigenvalues by activating sparser neurons (on ImageNet), especially in the first few layers.”**
> We find that SAM reduces the Hessian eigenvalues and also increases sparsity. According to Eq (3) and (4), such sparsity contributes to a reduced Hessian eigenvalue as it is closely related to the value of $B_k$. We modify this sentence in the revised paper to the following to prevent it from being misleading:
> “By analyzing some intrinsic model properties, we observe that SAM increases the sparsity of active neurons (especially for the first few layers), which contribute to the reduced Hessian eigenvalues”.

---

> > ### Author Response · Authors · 2021-11-16
> > **Responses to Reviewer 1uek - Continue**
> >
> > - **“The weight norms increase, implying the commonly used weight decay may not be an effective regularization alone.”**
> > We vary the strength of weight decay for ViT-B/16 and report the results in Appendix J of the revision. When training without SAM, weight decay (in a certain range) helps improve the accuracy on ImageNet, and the weight norm also decreases when we enlarge the decay strength as expected. However, enlarging the weight decay aggravates the problem of converging to a sharper region measured by both $L_{train}^\mathcal{N}$ and $\lambda_{max}$. We observe that $\|w\|_2$ consistently increases after applying SAM for every weight decay strength, together with the improved ImageNet accuracy and smoother landscape curvature.
> >
> > - **“A side observation is that, unlike ResNets and MLP-Mixers, ViTs have extremely sparse active neurons, revealing the redundancy of input image patches and the capacity for network pruning.” and the sparsity of Mixer activations “suggests the potential redundancy of image patches.”**
> > As the active neurons are sparse in the forward pass (see Figure 2 in the revision, the portion of active neurons are pretty low for ViT), there exist many neurons that make minor contributions to the final prediction. In fact, there are papers showing that the final prediction in ViTs is only based on a subset of image patches [5], and one can safely drop a large portion of tokens in Transformers of the vision domain [6].
> >
> > We address other comments below:
> >
> > - **It would be interesting to see whether SAM allows vision transformers to be trained with SGD.**
> > We test whether SAM can allow ViTs to be trained with SGD. In accordance with previous analysis [1, 2], (momentum) SGD lays far behind Adam when training Transformers. The ViT-B/16 trained with SGD only achieves 71.5\% accuracy on ImageNet, which is much worse than Adam (74.6\%). We surprisingly find that SGD+SAM can tremendously boost its accuracy to 79.1\%, which is a 7.6\% absolute accuracy increase. This result shows that SAM is able to allow ViTs to be effectively trained with SGD. We include this result in our revision.
> >
> > - **Does Figure 2a depict training with or without SAM? I think it would be informative to plot both to depict the effect of SAM over the course of training.**
> > Figure 2 (left) shows the learning curves without SAM, and Figure 2 (middle) shows the effects of SAM and strong augmentations. We can see that although SAM increases the training error, it improves the generalization (test accuracy) on ImageNet.
> >
> > - **I think the authors should extend their analysis and assess the average flatness of other models with vs. w/o SAM.**
> > We extend this analysis and include the results for all models in Table 1 of the revised paper. SAM can smooth the average flatness for all three architectures.
> >
> > - **In Section 4.2. Better Robustness, the effects of SAM on robustness in ResNets should be mentioned. Furthermore, percent changes in evaluation metrics such as accuracy must be specified as relative or absolute (i.e. percentage point). Finally, the authors need to control for the effect of the clean accuracy improvement: is the change in ImageNet-C accuracy larger than expected given the baseline change in ImageNet accuracy?**
> > We include the effects of SAM on ResNets’ robustness in the revision. Note that, if not specifically pointed out, the improvement stated in our paper denotes the absolute accuracy increase. We also add some discussions in Section 4.2, revealing that SAM improves the model robustness on ImageNet-C more than the clean accuracy on ImageNet considering the relative improvement. For ViT-B/16, Mixer-B/16, and ResNet-152, their relative improvements on ImageNet brought by SAM are 7.1%, 16.6%, and 1.0%, whereas the relative improvements on ImageNet-C are larger: 21.2%, 44.4%, and 4.4%, respectively.
> >
> > - **All the plots in Figure 2 should also contain a ResNet-152, with and without SAM.**
> > The loss function for optimizing ViTs and Mixers are the sigmoid loss in Dosovitskiy et al. [3] and Tolstikhin et al. [4], while the loss function for ResNets are the softmax loss, so their loss values are not totally comparable. We also clarify this in the Appendices.
> >
> > - **I understand that there are space constraints, but the ViT sparsity results should be depicted in a figure.**
> > We update Figure 2 (right) to include the sparsity results for all three architectures. It shows that the proportion of active neurons in ViT is much smaller than the other two architectures. SAM makes the first few layers more sparse for Mixer-B/16.
> >
> > - **The presentation of the contrastive learning results (Section 5.2) should include results for the baseline, ResNet-152.**
> > We include the results in the revision, ResNet-152 improves from 79.7% to 80.0% after enhanced by SAM.
> >
> > - **Figure 4a should show values both with and without SAM.**
> > We update the figure and include all the results in the revision.

---

> > > ### Author Response · Authors · 2021-11-16
> > > **Responses to Reviewer 1uek - Continue**
> > >
> > > - **The authors should include a discussion of limitations of their work.**
> > > We include a discussion of limitations in the revised paper (Conclusions and Limitations).
> > >
> > > [1] Zhang et al. Why are Adaptive Methods Good for Attention Models? NeurIPS 2020.
> > > [2] Liu et al. Understanding the Difficulty of Training Transformers. EMNLP 2020.
> > > [3] Dosovitskiy et al. An Image is Worth 16x16 Words: Transformers for Image Recognition at Scale. ICLR 2021.
> > > [4] Tolstikhin et al. MLP-Mixer: An all-MLP Architecture for Vision. 2021.
> > > [5] Rao et al. Efficient Vision Transformers with Dynamic Token Sparsification. NeurIPS 2021.
> > > [6] Akbari et al. VATT: Transformers for Multimodal Self-Supervised Learning from Raw Video, Audio and Text. 2021.

---

> > > > ### Comment · Reviewer_1uek · 2021-11-22
> > > > **Response to rebuttal**
> > > >
> > > > Thank you for your detailed response. The new results and explanations make the work a lot more convincing. There are a few minor issues to be resolved, but the paper is otherwise solid. Unless other reviewers are able to convince me of a fatal flaw in this work, I will recommend it be accepted, and have updated my score from a 3 to an 8 accordingly.
> > > >
> > > > **Remaining concerns:**
> > > >
> > > > What’s the difference between the two hybrid model variants (Table 3)?
> > > >
> > > > I think the claims about attention map interpretability still need to be softened. I would suggest changing “Another interesting finding is that ViTs’ performance gain also translates to visually more interpretable attention maps” to something like “Another interesting finding is that the improved ViT performance appears to change the quality of its attention maps“
> > > >
> > > > >All the plots in Figure 2 should also contain a ResNet-152, with and without SAM.
> > > >
> > > > If the losses in Figure 2 are not comparable then the test and validation accuracies can be plotted.

---

> > > > > ### Author Response · Authors · 2021-11-24
> > > > > **Thanks a lot for your response!**
> > > > >
> > > > > Following the exact hybrid model architectures in Dosovitskiy et al. [1], both R50-S/16 and R50-B/16 take the feature map of a ResNet-50 as the input sequence. Their difference is that R50-S/16 uses the ViT-S/16 model to process the sequence, while R50-B/16 uses the ViT-B/16. We train the full models from scratch, meaning that the ResNet-50 is randomly initialized.
> > > > >
> > > > > For the attention map interpretability and Figure 2, we will improve the paper per your advice (sorry that OpenReview does not allow us to update the submission anymore; Nov 22nd was the deadline for paper revision.
> > > > >
> > > > > [1] Dosovitskiy et al. An Image is Worth 16x16 Words: Transformers for Image Recognition at Scale. ICLR 2021.

---

### Official Review · Reviewer_hGsf · 2021-11-05

**Correctness:** 3
**Technical Novelty And Significance:** 3
**Empirical Novelty And Significance:** 3
**Recommendation:** 6
**Confidence:** 4

**Main Review:**

The main contributions of the paper are: 1 ViTs with existing sharpness-aware minimizer (SAM) outperform ResNets of similar size and throughput when trained from scratch on ImageNet without large-scale pre-training or strong data augmentations. 2 The motivation and experiments, including the ablation studies are well-organized and adequate. The paper reveals that the ViTs and MLP-Mixers have extremely sharp local minima of converged models through loss landscape visualization and Hessian dominate value.

The impact of the paper may be limited. The paper only compares with ResNet-152 with no augmentation and large dataset pre-training. Currently, the augmentation and large dataset pre-training are vastly employed in the entire community. The impact can be enlarged by conducting experiments on ViT-L and comparing ResNet152x4 pretrained on BiT-L.

**Summary Of The Paper:**

The paper is clearly written with extensive experiments. The paper compares dataset size, data augmentation, and different network architectures, ViTs, ResNets and MLP-Mixers by using loss landscape visualization, number of parameters, neural tangent kernel (NTK), Hessian matrix, training dynamics, percentage of activated neurons, throughput, norm of weights and activations, attention map visualizations and missing rate under linear interpolation.

**Summary Of The Review:**

The paper is pretty well-written with extensive results and figures. The impact of the paper can be largely improved if the large-scale experiments can be conducted and a better accuracy than the current state-of-the-art methods can be achieved.

---

> ### Author Response · Authors · 2021-11-16
> **Responses to Reviewer hGsf**
>
> Thanks for your constructive and positive feedback, below are our responses regarding your concerns.
>
> - **The paper only compares with ResNet-152 with no augmentation and large dataset pre-training. Currently, the augmentation and large dataset pre-training are vastly employed in the entire community.**
> We compare ViT with ResNet-152 under strong data augmentations in Section 4.5. ViT-B/16-AUG achieves 79.6\% accuracy, outperforming ResNet-152-AUG (78.8\%). Section 5.1 shows that SAM can lead to improvement for all three architecture species when they are pre-trained on ImageNet-21k. We emphasize the improvement on pre-training in the revision.
> Large-scale datasets are hard to collect, and strong augmentations incur many additional hyperparameters. Therefore, it is still important to study the inherent properties of various architectures and their behaviors without pre-training and augmentations. We identify a battery of tools for analyses and a principled optimization algorithm to greatly improve convolution-free architectures in such circumstances. Hence, we believe that our analyses can have positive impacts on the community.
>
> - **The impact can be enlarged by conducting experiments on ViT-L and comparing ResNet152x4 pretrained on BiT-L.**
> We train a ViT-L/16 (304M parameters) on ImageNet to further validate the effects of SAM. Its accuracy on ImageNet improves from 74.0% (vanilla training) to 80.8\% (+6.8\%). This result supports our claim in the paper that SAM helps larger models even more than smaller ones (e.g., compared +6.8\% for ViT-L/16 with +5.3\% for ViT-B/16 and +3.7\% for ViT-S/16). ResNet-152x4 has 938M parameters, so it is hard for us to train due to the memory constraint. For your reference, ResNet-152x4 with 480 image resolution (while we use 224 throughout our paper) achieves 81.3\% accuracy [1] when trained from scratch on ImageNet (without SAM).
>
> [1] Kolesnikov et al. Big Transfer (BiT): General Visual Representation Learning. ECCV 2020.

---

### Official Review · Reviewer_4nyX · 2021-11-25

**Correctness:** 3
**Technical Novelty And Significance:** 4
**Empirical Novelty And Significance:** 4
**Recommendation:** 8
**Confidence:** 4

**Main Review:**

The paper is experimental, non-theoretical kind, mostly studyng the effect of SAM on ResNet, ViT and MLP-Mixer training and its interplay with dataset size and augmentation. Also paper studies how SAM influences different aspects of the trained network, namely - attention masks, eigenvalues of the Hessian of models blcoks and loss landscape.

The main stated result is that SAM can more or less replace the heavy image augmentations for ViT and MLP-Mixer (Result1) and improve accuracy on clean and corrupted test set.

Strong points:

- Impact of the Result1. I think, that despite of the paper simplicity (let’s apply SAM to the non-CNN architectures), the message that one can replace (domain-specific) augmentations with (general-purpose) optimizer is of great importance. All sota methods for image classification rely on the augmentations  to provide good results and it is not easy to come up with new augmentations. The most papers, which present “learned augmentations” in fact just learn some parameters and combinations of _known_ augmentation. However, for other domains, like NLP, or some kind of medical data, it is not clear, how one could augment the data. If one can instead just take general-purpose architecture (transformer) and train it with general-purpose optimizer (SAM), that is a big deal.
- The experiments are done rigorously and with care. The only possibly missing experiment would be to train a model without an inductive bias at all, like vanilla MLP and check if SAM would greatly improve its results or not.
- The paper also states the limitations of the approach, the main of which is computational cost

Weak points:

- In the current form (I was asked to review a paper after the rebuttal stage), I do not see any major weaknesses.

**Summary Of The Paper:**

The paper presents an application of sharpness-aware minimization (SAM) to training of visual transformers (ViT) and MLP-Mixer. The idea of the paper is that models with weaker inductive priors (transformers and MLP-Mixer) suffer from sharp local minima more, than CNN-models and SAM fixes this issue, also significantly improving test accuracy on various version of ImageNet.

**Summary Of The Review:**

The paper tackles a single problem and studies it extensively. The problem in my opinion is significant and the paper would likely have an impact. Therefore I vote for clear acceptance.

---

> ### Author Response · Authors · 2021-11-30
> **Response to Reviewer 4nyX**
>
> We appreciate your effort and thanks a lot for your positive feedback!

---

### Author Response · Authors · 2021-11-16
**Revision Summary**

We thank all reviewers for the careful and constructive reviews. We have revised the paper accordingly and marked the newly added materials and modifications in blue color for visibility. The changes are as follows.

- We add results for a hybrid architecture (ResNet+ViT) to further show that the degree of improvement brought by SAM negatively correlates with the constraints of inductive biases built into the architecture. The hybrid model achieves 82.4\% accuracy on ImageNet without any strong augmentations and pre-training.
- We extend the analysis for average loss geometry to all three architectures. SAM can smooth both the average and the worst-case landscapes.
- We add the results for ViT when removing the softmax constraint. The resultant architecture is more prone to overfitting and converging to sharp regions than the original ViT. This result helps understand the difference between ViT and Mixer.
- We add the experiments for (momentum) SGD + SAM. While vanilla SGD performs much worse than Adam (71.5\% vs. 74.6\%), SAM greatly reduces their gap. SGD + SAM can train a ViT-B/16 that achieves 79.1\% accuracy on ImageNet.
- We vary the strength of weight decay to show that it is not effective in avoiding the convergence to sharp regions.
- We update Figure 2 to include the sparsity results for all architectures. ViT is sparser than the other two. Figure 4 now includes the result without SAM.
- We move the adversarial and contrastive learning section to Appendix due to the space constraint. We also include the contrastive learning result for ResNet-152.

---

### Public Comment · ~Yao_Fu3 · 2022-04-02
**A highly-related concurrent work**

Dear authors,

Thank you for this great work! The results are quite inspiring and there are multiple intriguing points from the optimization perspective. Just like to point out a concurrent paper that shares very similar inspiration with this work (with slightly more focus on the theoretical side and empirical improvements over SAM): https://openreview.net/forum?id=edONMAnhLu- (also on ICLR)
Hope it would be of interest!

---

> ### Public Comment · ~Boqing_Gong1 · 2022-04-19
> **Thank you for the pointer to "Surrogate Gap Minimization Improves Sharpness-Aware Training"**
>
> Dear Yao, Thank you for the great words about our work and the pointer to https://openreview.net/forum?id=edONMAnhLu- (Surrogate Gap Minimization Improves Sharpness-Aware Training), my other paper with collaborators. Here is the story behind the two papers: We tried to improve SAM's computational efficiency after noticing its superb performance boost to ViT. We didn't find a satisfying solution to the computation efficiency... but we gained a much better understanding of SAM from the failed trials, which led us to the new optimizer, i.e., the surrogate gap guided SAM (GSAM). SAM's or GSAM's computational efficiency remains a research challenge/opportunity.

---

### Decision · Program_Chairs · 2022-01-20

**Decision:**

Accept (Spotlight)

**Comment:**

### Description
The paper demonstrates that efficient architectures such as transformers and MLP-mixers, which do not utilize translational equivariance in the design, when regularized with SAM (sharpness aware minimization) can achieve same or better performance as convolutional networks, in the vision problems where the convolutional networks were traditionally superior (with data augmentation or not, regularized or not). The paper demonstrates it very thoroughly through many experiments and analysis of the loss surfaces.

### Decision + Discussion

I find the paper to be very timely in its context. It has a remarkable coverage of experimental studies and different use cases: SAM + augmentation, +contrastive, +adversarial, +transfer learning; as well as ablation studies such as keeping first layers convolutional. The reviewers have asked further questions, and the authors were able to conduct respective experiments within the discussion period fully addressing all concerns and making the findings of the paper even more comprehensive and convincing.

After the rebuttal 3 reviewers were for "accept", one "marginally above" and one "marginally below". In the latter case the concern was that the paper is an experimental study of a known method, SAM. While I understand that many researchers are expecting theoretical and innovative results from ICLR papers, I find that it does not prevent acceptance. Indeed, the experimental findings in this paper are on a "hot" topic, could be of wide interest and could lead to a change of paradigm in designing models towards more generic ones. On the other hand, it could just indicate that CNNs are not fully exploiting their potential, e.g. not exploiting the context well enough in the hidden layers?

To get more insight, I am still wondering, how the predictions behave if the input is shifted by a few pixels in CNN and Transformers? It seems counterintuitive that making the first layers in ViT just an MLP of image patches is a good design. Furthermore, fully convolutional models allow to take input of an arbitrary size and average the predictions on the output if it happened to be larger than 1x1.

Since convolutions are also used for e.g. semantic segmentation and generative models, one should not (and the authors do not in the paper) discard them too fast. See also a recent work combining transformers and convolutional networks,
Chen et al. (ICCV 2021) Visformer: The Vision-friendly Transformer.

---

> ### Public Comment · ~Xiangning_Chen1 · 2022-03-16
> **Reply to Program Chairs**
>
> Thanks a lot for your positive comment and constructive feedback!
>
> Regarding your question, we find that the accuracy of ViT won't drop a lot if we shift/roll the pixel.
> We use the vanilla ViT-B/16 with 74.6% accuracy and input resolution as 224. Please see the table below for the detailed results.
>
> | Roll Direction 	| Roll Pixel 	| Accuracy (%) 	|
> |:--------------:	|:----------:	|:------------:	|
> |     Height     	|     10     	|     73.9     	|
> |      Width     	|     10     	|     74.1     	|
> |      Heigh     	|     30     	|     72.9     	|
> |      Width     	|     30     	|     73.6     	|
> |     Height     	|     100    	|     70.1     	|
> |      Width     	|     100    	|     72.4     	|
>
> It's probably because that during training, the image is already preprocessed by random crop + horizontal flipping, so the model is not very sensitive to the image shift.